# Universal One-third Time Scaling in Learning Peaked Distributions

Yizhou Liu [1]   Ziming Liu [1 2]   Cengiz Pehlevan [3]   Jeff Gore [1]

## Abstract

Training large language models (LLMs) is computationally expensive, partly because the loss exhibits slow power-law convergence whose origin remains debatable. Through systematic analysis of toy models and empirical evaluation of LLMs, we show that this behavior can arise intrinsically from the use of softmax and cross-entropy. When learning peaked probability distributions, e.g., next-token distributions, these components generically yield power-law vanishing losses and gradients, regardless of many microscopic details, creating a fundamental optimization bottleneck. This ultimately leads to power-law time scaling of the loss with a universal exponent of $1/3$. Our results provide a mechanistic explanation for observed neural scaling and suggest new directions for improving LLM training efficiency.[1]

## 1. Introduction

Neural scaling laws describe how the pre-training loss of large language models (LLMs) decreases with model size and training dataset size according to power laws, enabling performance predictions for larger models trained on more data (Hestness et al., 2017; Kaplan et al., 2020; Rae et al., 2021; Hoffmann et al., 2022). Despite the accuracy and predictive power of these fitted power laws, they remain largely empirical. Key questions, e.g., whether loss scaling can be made more efficient and when these scaling laws will break down, remain unanswered. Addressing these questions requires understanding the origins of the observed power laws.

In this work, we focus on loss scaling with training time or dataset size. In the standard pre-training regime with fixed batch size and a single pass over data, training time is

proportional to the effective dataset size, so time-based and data-based power laws are equivalent (Michaud et al., 2023; Bordelon et al., 2024; 2025a;b). We therefore ask:

> **Q: Why power law and what exponent?**
>
> What mechanism **relevant to LLMs** gives rise to a power-law decay of the loss over time? What determines the power-law exponent?

Many mechanisms have been proposed for neural scaling laws. Existing theories, spanning various mathematical frameworks, ultimately attribute power-law loss scaling to power-law structures in the data (Spigler et al., 2020; Hutter, 2021; Maloney et al., 2022; Sharma & Kaplan, 2022; Michaud et al., 2023; Bahri et al., 2024; Bordelon et al., 2025a;b). The intuition is that data contains various features, skills, or tasks to be learned. If these entities have power-law importance or frequency distributions, and the model learns the more important or frequent ones first, then loss will decay as a power law with training time. The exponent then depends on the power law in the data structure.

Here, we identify another mechanism not sensitive to data but more related to the architecture. We find that **softmax function** (Boltzmann distribution) and **cross-entropy loss** together lead to power-law loss and gradients when learning peaked distributions (low-entropy distributions). This leads to power-law loss decay with training time having exponent $1/3$. Crucially, deriving this exponent requires only Taylor expansion and time integration, i.e., the result is independent of data properties beyond the distributions being peaked. This mirrors the concept of universality in statistical physics (Kardar, 2007), where critical exponents depend only on broad system properties, not microscopic details.

To establish this mechanistic explanation for neural scaling, we identify minimal model components required to reproduce power-law training dynamics, arriving at a toy model that characterizes the language model (LM) head (Section 2). We next systematically analyze the identified toy model with theory and experiments, and conclude the universal $1/3$ time scaling of loss specifically due to properties of softmax and cross-entropy in learning peaked distributions (Section 3). To further justify the relevance of this mechanism to LLM training, we evaluate the Pythia models (Biderman et al.,

---

[1]Massachusetts Institute of Technology [2]Stanford University [3]Harvard University. Correspondence to: Yizhou Liu <liuyz@mit.edu>, Jeff Gore <gore@mit.edu>.

*Proceedings of the 43rd International Conference on Machine Learning*, Seoul, South Korea. PMLR 306, 2026. Copyright 2026 by the author(s).

[1]Code available at github.com/liuyz0/TimeScaling

2023) at different stages during training, and find that the next-token distributions are sufficiently peaked and the loss does obey $1/3$ time scaling (Section 4). We compare our findings to previous works in Section 5, and summarize our results with a discussion of limitations and implications in Section 6.

> **Our contributions are the following:**
>
> - We find that softmax and cross-entropy can lead to power-law training without power laws in data.
> - We show power laws arise intrinsically from the non-linearities when learning peaked distributions. Scaling exponent of loss versus time is then $1/3$.
> - We find evidence of this mechanism in LLMs.

Overall, our work is the first theoretical study showing that softmax and cross-entropy can be the key to power-law training behaviors. We emphasize the overlooked yet special and important roles of these non-linearities in shaping LLM training dynamics.

## 2. Toy Model

We try to find the minimum ingredients related to LLMs that can reproduce power-law training behaviors. By isolating different parts of LLMs, the minimum setup we identified is a one-layer network with softmax as non-linearity and cross-entropy as the loss function, mimicking the LM head.

We define our model using a teacher-student setup where both networks share the same architecture, and the student is trained to match the teacher's output. The teacher has a fixed weight matrix $W^* \in \mathbb{R}^{n \times m}$, while the student has trainable weights $W \in \mathbb{R}^{n \times m}$, where $m$ denotes the hidden dimension and $n$ the output dimension (number of classes). Given input (or hidden state) $x \in \mathbb{R}^m$, the teacher outputs a probability distribution

$$p(x) = \text{Softmax}(W^*x) \in \mathbb{R}^n, \tag{1}$$

where for a vector $v$, the $i$-th element of $\text{Softmax}(v)$ is

$$\text{Softmax}(v)_i = \frac{e^{v_i}}{\sum_j e^{v_j}}. \tag{2}$$

Similarly, the student outputs

$$q(x) = \text{Softmax}(Wx). \tag{3}$$

Kullback–Leibler (KL) divergence between teacher and student outputs gives loss on a single input $x$

$$L(x) = \sum_{i=1}^{n} p_i(x) \ln \frac{p_i(x)}{q_i(x)}. \tag{4}$$

KL-divergence and cross-entropy have a constant difference, the entropy of $p$, which leads to no difference in student

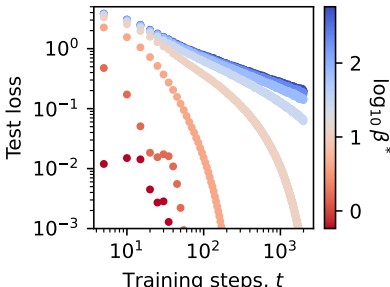

*Figure 1.* The toy model exhibits power-law training behaviors at low temperatures. At high temperatures (small $\beta^*$), the loss vs. step $t$ curves are concave on the log-log plot, similar to exponential decay. At low temperatures (large $\beta^*$), the loss vs. step $t$ curves converge to a line on the log-log plot, like power laws with the same exponent. Details in Section D.1.

gradients, student dynamics, or the convergence behavior (how the loss converges to its minimum value). We thus do not distinguish the two loss functions in our analysis. We use the KL-divergence as its minimum is zero, making plots of loss directly show convergence.

As a part of the setup, we sample the input or hidden state $x \in \mathbb{R}^m$ as i.i.d. standard normal. Then, we can define the (overall) loss as

$$L = \langle L(x) \rangle_x, \tag{5}$$

where $\langle \cdot \rangle_x$ denotes an average over the distribution of $x$. The teacher weight $W^* \in \mathbb{R}^{n \times m}$ is given by

$$W^* = \frac{1}{\sqrt{m}} \hat{W} \beta^*, \tag{6}$$

where entries in $\hat{W} \in \mathbb{R}^{n \times m}$ are i.i.d. standard normal, and $\beta^*$ is a scalar called **inverse temperature**. So far, we have introduced the toy model. For later convenience, we also introduce and emphasize two important points:

> **Key concepts**
>
> - Logits are $y_i^* = \sum_j W_{ij}^* x_j$ for teacher and $y_i = \sum_j W_{ij} x_j$ for student.
> - Inverse temperature $\beta^*$ is the standard deviation of the teacher logits, controlling how peaked $p$ is.

The inverse temperature $\beta^*$ is the most important hyperparameter controlling data properties in our toy model. Intuitively, a large $\beta^*$ produces a large logit variation, causing the distribution $p$ to concentrate on the largest logits (peaked distribution). A small $\beta^*$ makes all logits similar, yielding a uniform (flat) distribution $p$. We will infer inverse temperature from the logit standard deviation in our experiments.

We next explore training behaviors in the toy model by scanning inverse temperature $\beta^*$ over three orders of magnitude

($\sim 0.6$ to $\sim 600$) with fixed $m = 32$, $n = 128$, and a batch size 1024 (large enough to reduce noise). The student weight was initialized at zero and trained by Adam (Kingma & Ba, 2014) with a fixed learning rate (details in Section D.1). At high temperatures (small $\beta^*$), the loss decreases rapidly with training step $t$, resembling exponential decay (Figure 1). However, at low temperatures (large $\beta^*$), the loss versus $t$ becomes linear on a log-log plot, i.e., a hallmark of power-law scaling, with the exponent converging to a fixed constant (Figure 1). We therefore conclude that

> **Result 1: Toy model relevant to power-law training**
>
> Power-law loss with training time emerges in the toy model at low teacher temperatures.

The toy model captures the LM head: the inputs $x$ represent the final hidden states, and each output is a distribution over the vocabulary for next-token prediction. Having shown that (1) the toy model relates to LLMs and (2) the toy model exhibits power-law training, we next try to demonstrate that (3) the mechanism producing power laws in the toy model is relevant to, or dominates in, LLMs. This requires analyzing the toy model in depth and validating theoretical predictions on actual LLMs.

## 3. Toy Model Analysis

We analyze student weight dynamics via continuous-time gradient flow:
$$\frac{dW}{d\tau} = -c_{\text{eff}} \nabla L, \quad (7)$$

where $\tau$ is the time of optimization dynamics and $c_{\text{eff}}$ is a constant introduced for generality. In reality, dynamic time $\tau$ at the training step $t$ is $\tau = \int_0^t \eta_{t'} dt'$ where $\eta_{t'}$ is the learning rate at step $t'$. With a constant learning rate, $\tau \propto t$, so we use them interchangeably when studying scaling behavior. The constant $c_{\text{eff}} = 1$ for gradient descent; for advanced optimizers like Adam, $c_{\text{eff}} \neq 1$ captures the effective step size being different from the learning rate even when the trajectory can follow gradient direction on average (Section A.1).

Following the definition of loss, we can derive that
$$\frac{dW}{d\tau} = c_{\text{eff}} \langle (p - q) x^T \rangle_x. \quad (8)$$

Such dynamics are not analytically solvable in general, given the complex non-linearity.

We looked into the student weights during training for inspiration. The visualization of weight matrices (Figure 2) suggests that student weight $W$ is approximately proportional to teacher weight $W^*$, and the variable that changes during training is the norm of $W$. We therefore propose:

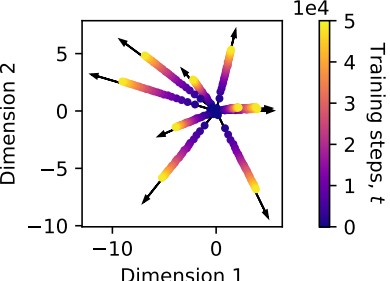

*Figure 2.* Actual training inspires the aligned student ansatz (student weight is proportional to the teacher's). We project rows of teacher weight $W_i^* \in \mathbb{R}^m$ and the corresponding student weight rows $W_i$ to a 2-dimensional space. The student weight is initialized at zero. The arrows are fixed teacher weight rows. The colored dots are student rows at different steps. Details in Section D.1.

> **Aligned student ansatz**
>
> The student's weight is proportional to the teacher's. The student weight is then $W(t) = \frac{1}{\sqrt{m}} \hat{W} \beta(t)$, where $\beta(t)$ is the student inverse temperature.

Heuristically, the aligned student ansatz can be approximately right throughout due to small initialization, which is preferred in practice as it promotes feature learning and parameter rotation (Yang & Hu, 2021; Atanasov et al., 2021; Kunin et al., 2024). The same update can lead to a larger change in angle for vectors with smaller norms. The student weight can easily rotate to the right direction at the early stage and focus on growing then.

With the ansatz, we can obtain the typical dynamics of $\beta$ from Equation (8) as (Section A.2)
$$\frac{d\beta}{d\tau} = -\frac{c_{\text{eff}}}{n} \frac{dL(\beta)}{d\beta}, \quad (9)$$

where $L(\beta)$ is the loss averaged over data and teacher randomness, satisfying
$$L(\beta) = U(\beta^*)(\beta - \beta^*) + \beta^* F(\beta^*) - \beta F(\beta), \quad (10)$$

with a purely mathematical quantity $U$, which we call "internal energy" by analogy, given by
$$U(\beta) = \frac{\partial(\beta F)}{\partial \beta}(\beta), \quad (11)$$

and the "free energy" $F$ defined from
$$\langle \ln Z \rangle = -\beta F(\beta). \quad (12)$$

Here, $Z = \sum_i e^{-\beta \epsilon_i} = \sum_i e^{y_i}$ is called the partition function with $\epsilon_i = -\frac{1}{\sqrt{m}} \sum_j \hat{W}_{ij} x_j$ called the energy associate

with class $i$ given input $x$. The symbol $\langle \cdot \rangle$ is short for $\langle \cdot \rangle_{\hat{W},x}$, meaning average over teacher randomness and data.

The key now is to understand the free energy $F$. We first consider the low temperature (large $\beta$) limit. At zero temperature, free energy collapses to the minimum energy $\langle \min_i \epsilon_i \rangle$. In our setup, $\epsilon_i$ are i.i.d. standard normal, and by extreme value distribution, $\langle \min_i \epsilon_i \rangle = -c_0$ where $c_0 \approx \sqrt{2 \ln n}$ (for large $n$) is a constant. Near zero temperature, the free energy can be expanded as a function of temperature $\beta^{-1}$, which is called the low-temperature expansion, for generic distributions of energies $\epsilon_i$ (Section A.3):

$$F(\beta) = -c_0 - c_1 \beta^{-1} - c_2 \beta^{-2} + \cdots . \quad (13)$$

We therefore have

$$U(\beta) = -c_0 + c_2 \beta^{-2} + \cdots . \quad (14)$$

When both student and teacher are in the low temperature regime (we can apply the above expansion to both $\beta$ and $\beta^*$), we have the loss

$$L = c_2(\beta^{-1} - \beta^{*-1}) + c_2(\beta - \beta^*)\beta^{*-2} + \cdots , \quad (15)$$

and the negative gradient

$$-\frac{dL}{d\beta} = U(\beta) - U(\beta^*) = c_2(\beta^{-2} - \beta^{*-2}) + \cdots . \quad (16)$$

In the **intermediate regime** where $\beta$ is large enough while much smaller than $\beta^*$, we have the leading behaviors

$$L \approx c_2 \beta^{-1}, \quad -\frac{dL}{d\beta} \approx c_2 \beta^{-2}, \quad (17)$$

subsequently, according to Equation (9),

$$\beta \sim \tau^{1/3}, \quad (18)$$

and finally

$$L \sim \beta^{-1} \sim \tau^{-1/3}. \quad (19)$$

For high temperature data (small $\beta^*$), it seems that the training dynamics is not power-law (Figure 1), which is confirmed by theory as well (Section A.4). In the high-temperature regime, $U(\beta) \approx -\beta$. Since we know $U(\beta) \geq -c_0$ for any $\beta$, the value $\beta = c_0$ may demarcate the boundary between high and low temperature regimes.

To test the theory, varying $\beta$ and $\beta^*$, we numerically sampled many $\hat{W}$ and $x$ to obtain loss and gradient under the aligned student ansatz (details in Section B.1). Both the loss $L$ and negative gradient $-dL/d\beta$ start to be power-law starting around $\beta \approx c_0$ and deviate from power laws when $\beta$ is close to $\beta^*$ (Figure 3). Increasing $\beta^*$, we obtain a clearer and wider power-law region, whose fitted exponents are

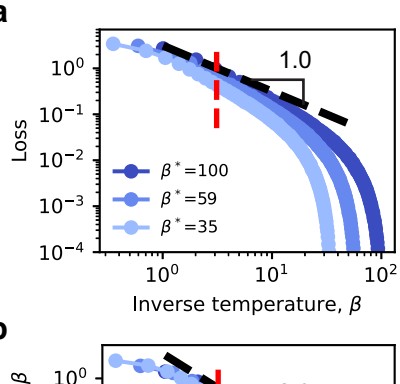

**a**

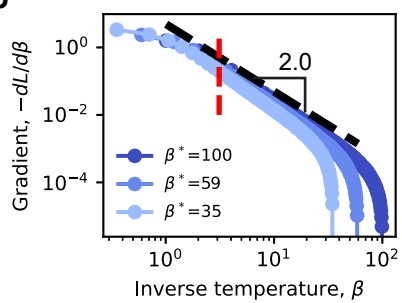

**b**

*Figure 3.* Numerical loss and gradient agree with the theory. The vertical dashed line is $\sqrt{2 \ln n} \approx c_0$. Both loss (panel a) and gradient (panel b) tend to be power-law with $\beta$ in the intermediate regime. With increasing $\beta^*$, the power-law regions are clearer, and the exponents seem to converge. Fitting of the curves under $\beta^* = 100$ yields that the loss exponent is $1.15 \pm 0.04$ and the gradient exponent is $2.26 \pm 0.01$. Details in Section B.1.

closer to our theory prediction. We therefore conclude that our theory is valid:

> **Result 2: 1/3 time scaling in aligned students**
>
> In the intermediate regime where $\beta$ is large yet much smaller than $\beta^*$, we have the loss $L \sim \tau^{-1/3}$ robust to other details.

The low-temperature expansion seems to be magic, and we seek to extract more intuitions from it. The origin of the power-law dynamics is deeply rooted in the use of the Boltzmann distribution (softmax) and the cross-entropy (or KL-divergence) loss. In such a system, to match a peaked distribution, we need a large $\beta$ (low temperature). Yet, in the low-temperature limit, both free energy and internal energy converge to the lowest energy smoothly with temperature, and the leading terms are therefore power-law with $\beta$. With the gradient flow dynamics, we inevitably obtain the power-law loss $L \sim \tau^{-1/3}$. The derivation of the exponent $1/3$ requires **Taylor expansion** and **time integration**, which is closely related to the idea of "universality" in statistical mechanics (Kardar, 2007). Other details, including many data structures, do not change the Taylor expansion argument or the exponent $1/3$, but may alter the coefficients $c_0$,

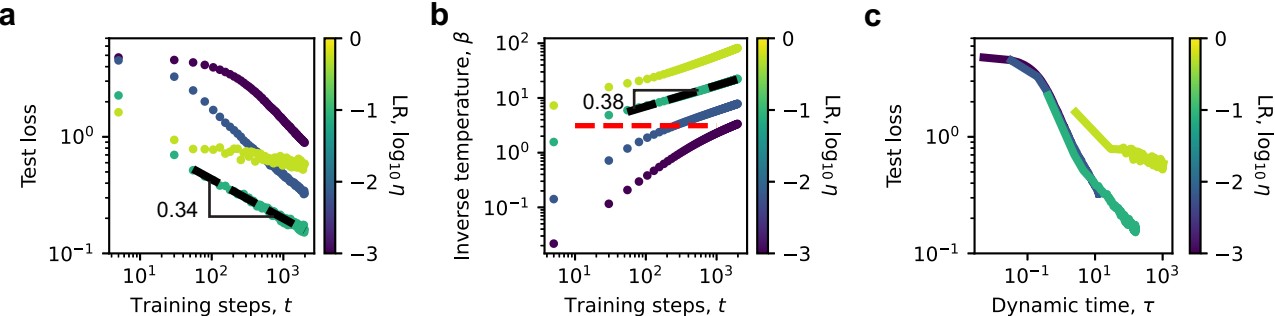

*Figure 4.* Aligned student ansatz can describe actual training results. Initializing student weights at zero and scanning learning rates $\eta$, we found that the loss under the optimal learning rate (lowest loss with the same step) has clear $1/3$ scaling (panel a), and the corresponding inverse temperature grows with an algebraic exponent 0.38 close to the theoretical value $1/3$ (panel b). Here, $\tau \propto t$, and power-law fitting with $t$ or $\tau$ does not change the exponent. The red dashed line means $\beta = c_0$. Larger learning rates cannot follow gradients to align the student. Smaller learning rates follow the same dynamics (gradient flow) whose curves collapse with the dynamic time $\tau$ (panel c), some of which do not show $1/3$ scaling clearly as their $\tau$ is small and are not deep in the low-temperature regime. More details in Section D.1.

$c_1$, $c_2$, etc. (e.g., $c_0 \approx \sqrt{2 \ln n}$ is specific to i.i.d. Gaussian logits). We use the word "universal" to echo "universality," which does not mean that the $1/3$ time scaling is true for all cases (only true in the intermediate regime), but that, in the intermediate regime, the exponent $1/3$ is generically true.

We next systematically study the actual training of the toy model. We want to understand how widely and accurately the aligned student ansatz can describe actual training, and the behavior of loss dynamics when our ansatz is wrong.

Given a large $\beta^*$, we initialized the student weight at zero and trained it using Adam with a fixed learning rate, but scanned different learning rates from $10^{-3}$ to 1. We found that under the optimal learning rate, which gives the minimum loss given the same number of steps, the loss follows $1/3$ time scaling (Figure 4a) and the corresponding inverse temperature $\beta$ measured from logit standard deviation also agrees with the theory (Figure 4b). A range of different $\beta^*$ were also scanned, and the phenomena are similar for large $\beta^*$ (Section D.1). Under small initialization, the aligned student ansatz can describe the actual optimal training.

We then try to understand the effect of learning rates. If the learning rate is too large, the loss is very noisy and non-decreasing (Figure 4a), while the inverse temperature is larger than that under the optimal learning rate (Figure 4b). This means that under too large learning rates, the student is not aligned with the teacher and has difficulty following the gradient to rotate. This may be explained by the fact that larger learning rates introduce more noise. For learning rates lower than the optimal one, it turns out that the student has smaller updates in total (Figure 4b) and has not entered the low temperature regime enough. If we plot the loss (Figure 4c) as a function of dynamic time $\tau$, the curves under learning rates smaller than the optimal learning rate collapse well, suggesting they obey the same dynamics –

possibly following the gradient flow. If we use SGD, the results agree with theory well: for all the learning rates we scanned, the curves can collapse to one when plotting with respect to $\tau$ (Section D.2). We also tested Adam with a learning rate scheduler, whose result agrees with the theory (Section D.4). To conclude, under small initialization and following gradient flow, actual optimization exhibits $1/3$-time scaling behaviors, agreeing with the aligned student ansatz theory.

Naturally, we next explore the effect of initialization scales. We initialize the student weight independent of teacher's and control the magnitude. The ratio $\beta_0/\beta^*$, where $\beta_0$ is the initial inverse temperature of the student, is also the ratio of the student weight norm and the teacher weight norm. We scanned a range of learning rates (Section D.3) and show the results from a sufficiently small learning rate (Figure 5, a and b). With an increasing initialization ratio $\beta_0/\beta^*$, the student cannot be aligned at the beginning, where the loss can have a sharp drop (Figure 5a), and $\beta$ can decrease first and then increase (Figure 5b). However, later, all the curves seem to collapse and agree with the aligned student ansatz predictions. We conclude that, given enough time, the student can be aligned, even if it is not initially, which leads to the $1/3$ time scaling.

Can $1/3$ time scaling occur when the aligned student ansatz fails? After adding weight decay (Loshchilov & Hutter, 2017), we found a totally new phenomenon. We set the weight decay to be non-negligible and also not to cause too much damage to performance or dynamics (Section D.5). For some cases, where the student can enter a low temperature regime and stay still far away from the teacher for a while, we observe $1/3$ time scaling of loss (Figure 5c). However, the corresponding inverse temperature $\beta$ is almost non-increasing and does not agree with the aligned student

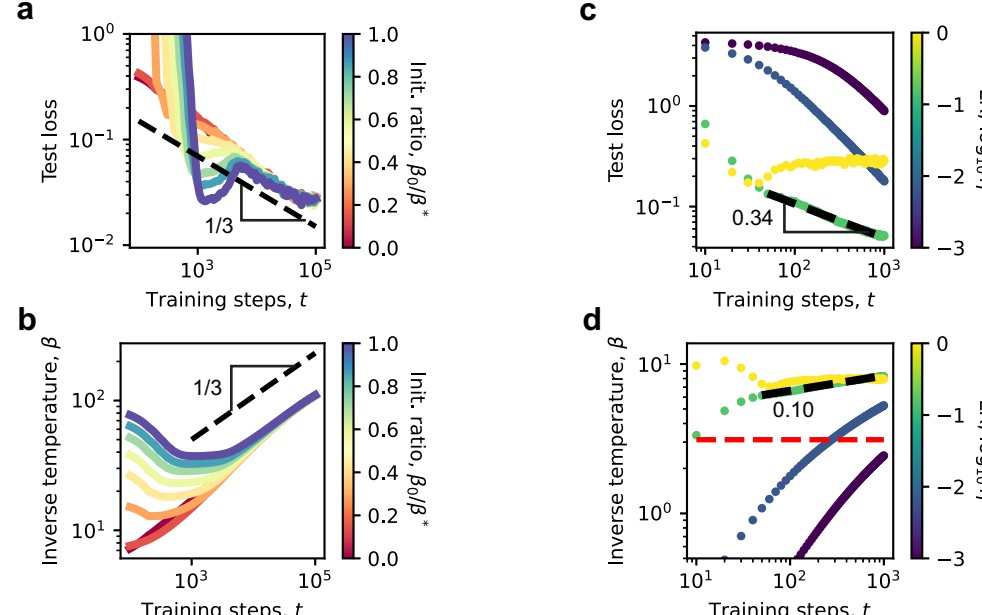

*Figure 5.* The $1/3$ time scaling of loss can be true beyond the aligned student ansatz. (a and b) Initializing the student at different scales, the aligned student ansatz is wrong at first, but can be correct later, yielding the $1/3$ time scaling. Details in Section D.3. (c and d) By adding weight decay, we observe that under the optimal learning rate, $1/3$ time scaling of loss is correct, while the aligned student ansatz is wrong since $\beta$ does not follow the theory. Beyond norm growth, rotation of parameters in the low-temperature regime may also lead to the $1/3$ time scaling of loss. Details in Section D.5.

ansatz (Figure 5d). The $1/3$ time scaling of loss then must be due to the rotation of the student weight rather than magnitude growth. If the student is too similar to the teacher, any dynamics can be approximated by local dynamics, which leads to an exponential convergence. If the student is too different from the teacher and has a large norm, as shown in Figure 5, a and b, the student weight can shrink and rotate, which leads to a much faster loss decay than the $1/3$ time scaling. The $1/3$ time scaling in Figure 5c due to rotation should then also be in an intermediate regime, where the student is similar to the teacher, as we initialize the student weight at zero, but is not too similar. We conclude, combining the analytic theory and experimental observations:

> **Result 3: $1/3$ time scaling in general students**
>
> In learning peaked distributions, when the student is similar to the teacher but not too close, we have $L \sim \tau^{-1/3}$, robust to other details.

The loss decay due to rotation may also exist in earlier experiments. Yet, we tune hyperparameters to make the effect of rotation dominate here. Though we cannot analytically solve dynamics when the student and teacher are misaligned, we offer a heuristic explanation. In the low-temperature regime, power-law vanishing loss and gradients likely occur for all parameter directions, not just the aligned direction.

This may produce the same power-law loss dynamics regardless of the specific direction of parameter updates.

To understand the generality of the $1/3$ time scaling beyond the current toy setup, we further studied the same toy architecture with power-law structures in the data (Section D.6), a generalized toy model with additional layers before the projection head (Section D.7), and a multi-layer perceptron (MLP) for (LeCun, 1998) MNIST classification (Section D.8). In all these cases, we observed the same $1/3$ loss scaling with training time. The empirical evidence and insights from the solvable model indicate that the $1/3$ scaling depends only on several generic properties (e.g., the use of softmax, a peaked output distribution, and logits approximately following gradient-flow dynamics), and is independent of many other details. Despite their enormous complexity compared to the models studied here, LLMs may therefore exhibit the same $1/3$ time scaling because the differences may not affect the relevant generic properties.

## 4. LLM Experiments

Having identified the $1/3$ time scaling mechanism, we now investigate its relevance to LLMs. We imagine the toy model mimics the LM head, motivating two questions: Are next-token distributions sufficiently peaked? Do LLMs exhibit $1/3$ time scaling in loss?

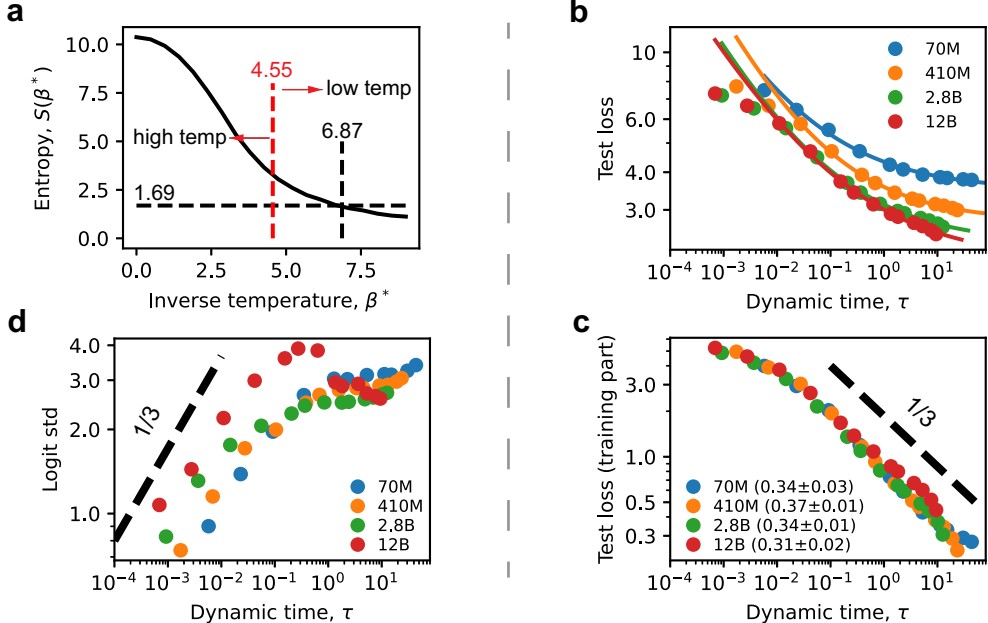

*Figure 6.* LLMs operate in the low-temperature regime and exhibit the $1/3$ time scaling behavior. (a) To reach the same entropy of next-token distributions, which is at most $1.69$ inferred from Chinchilla (Hoffmann et al., 2022), the i.i.d. Gaussian logits need to have a standard deviation $\beta^*$ at least $6.87$, which is larger than $c_0 \approx 4.55$ and falls into the low-temperature regime. (b) The raw loss $L$ (points) fitted by a power law plus a constant $L_{\setminus \tau}$ for each model. Different colors refer to different model sizes. (c) The loss related to training, $L - L_{\setminus \tau}$, follows $1/3$ time scaling. The values in parentheses are fitted $\alpha_\tau$, respectively. (d) Logit standard deviations in LLMs grow with an exponent close to $1/3$, indicating the entrance of the low-temperature regime, and then saturate, suggesting that later loss decay is due to rotation or alignment. Details in Section D.9.

One problem is that our rough boundary for high and low temperature regimes in terms of inverse temperature is $c_0$, which depends on the data structure. It is easy to estimate $c_0$ for i.i.d. Gaussian logits, while actual language has more complicated structures. The easiest solution we have is to imagine an equivalent system where logits are i.i.d. Gaussian that can produce the same peaked distributions (we match the entropy $S$ of the language distributions), and to study the inverse temperature quantitatively there. From the irreducible loss of cross-entropy loss (Hoffmann et al., 2022), we know that the upper bound for next-token distribution entropy is $1.69$. To reach this entropy with $n$ i.i.d. Gaussian logits where $n$ is the vocabulary size now, the numeric result suggests that the inverse temperature (standard deviation) $\beta^*$ to be $6.87$ (Section D.9), which is above $c_0 = \sqrt{2 \ln n} \approx 4.55$ of i.i.d. Gaussian logits (Figure 6a). We therefore conclude that, based on the toy model theory, the next-token prediction distribution is in the low-temperature regime.

If the target distribution to output is low-temperature or peaked, any well-trained LLMs should fall into the low-temperature regime and exhibit the $1/3$ time scaling in loss. We test this hypothesis on Pythia models (Biderman et al., 2023) whose checkpoints on different training steps are published. For each model, we evaluated it at different steps

on the FineWeb dataset (Penedo et al., 2024), and fitted the test loss $L$ with

$$L = \frac{c_\tau}{\tau^{\alpha_\tau}} + L_{\setminus \tau}, \tag{20}$$

where the dynamic time $\tau$ for different steps can be obtained via the learning rate scheduler, and $L_{\setminus \tau}$ is a constant containing loss related to model size and language entropy (Section D.9). Fitting of the raw losses is in Figure 6b, where we can see that the later stage of training can agree well with Equation (20). The early stage of training probably has a high temperature and exhibits different dynamics. To demonstrate the quality of the power-law fitting, we plot $L - L_{\setminus \tau}$ in Figure 6c. Surprisingly, the curves from different models collapse, suggesting that $\tau$ is the fundamental variable and that values of $c_\tau$ and $\alpha_\tau$ are constants across model sizes. Moreover, there is a clear power-law region, and the fitted exponents $\alpha_\tau$ are very close to $1/3$. All these findings are consistent with the hypothesis that LLMs are learning peaked distributions, falling into the low-temperature regime, and exhibit the $1/3$ time scaling.

The Chinchilla scaling laws (Hoffmann et al., 2022) assumed a power law with dataset size and had a fitted exponent of $0.28$. They used different maximum learning rates and learning rate schedulers when varying model sizes and dataset sizes, introducing a roughly proportional but essen-

tially non-linear relationship between the dataset size and the dynamic time $\tau$. Therefore, they can obtain an exponent $0.28$ not far away from $1/3$ but also not as close as our fitting. We emphasize that, in our theory, the loss is a power law essentially with the dynamic time $\tau$, not the dataset size.

We further study the logits of the LM head to gain more insights into LLM training dynamics. Note that since the logits of actual LLMs may not be i.i.d. Gaussian, the logit standard deviation (inverse temperature) values cannot be directly compared to the theory from our toy model, while the scaling behavior can be informative. We saw that the logit standard deviation slowly increases with an algebraic exponent close to $1/3$ at first (Figure 6d), which is the signature of the low-temperature regime and again supports the argument that LLM training enters the low-temperature regime. Later, the logit standard deviation stops growing while the loss scaling still follows $1/3$ time scaling (Figure 6c). This phenomenon is similar to the toy model experiment with weight decay (Figure 5, c and d). Indeed, Pythia models are trained with significant weight decays (Biderman et al., 2023). The loss decrease at the later stage is then mainly due to rotation of parameters rather than norm growth.

In addition to Pythia models (Biderman et al., 2023), we also evaluate Olmo models (OLMo et al., 2024), which show similar $1/3$ time scaling in loss and therefore support that the $1/3$ time scaling is general in LLMs (Section D.9). To summarize the section, we conclude:

> ### Result 4: $1/3$ time scaling in LLMs
>
> Next-token distributions are peaked, LLMs enter the low-temperature regime, and have the $1/3$ time scaling in loss, supporting our mechanism.

## 5. Related Works

We now compare our findings with previous works. Empirical studies on neural scaling laws demonstrated a power law with dataset size (Kaplan et al., 2020; Brown et al., 2020; Henighan et al., 2020; Hoffmann et al., 2022; Achiam et al., 2023). Following these observations, theoretical works with a "static" view studied the optimal test loss (as many training steps as possible) given the training dataset size. They argue that properties of data, like data manifold dimension (Spigler et al., 2020; Hutter, 2021; Sharma & Kaplan, 2022) and power-law spectrum in data covariance (Bordelon et al., 2020; Maloney et al., 2022; Bahri et al., 2024; Brill, 2024), can lead to a power-law loss with the dataset size.

Here, we followed a "dynamic" view closer to the reality of LLM training, which suggests that the origin of the power law is in training dynamics, and loss may be fitted as a power law with dataset size due to the use of online learning (each

step, LLMs see a batch of new data, and the dataset size is proportional to the number of steps). Previous works with this dynamic view did not study the critical role of softmax or cross-entropy. They either tried to solve toy models with MSE loss (Bordelon & Pehlevan, 2021; Lin et al., 2024; Bordelon et al., 2024; Worschech & Rosenow, 2024; Fonseca et al., 2024; Paquette et al., 2024; Bordelon et al., 2025a; Defilippis et al., 2025; Bordelon et al., 2025b) or applied high-level arguments (Michaud et al., 2023; Arora & Goyal, 2023; Liu et al., 2025d; Cagnetta et al., 2025) without analyzing the effect of non-linearity. Their conclusions are similar to those from the static view: the power-law loss with the dynamic time is due to data structures, e.g., certain assumed power-law spectrum (Bordelon et al., 2024; 2025a;b) or feature frequencies (Michaud et al., 2023). Our work found that softmax and cross-entropy can change this picture qualitatively, producing power laws by themselves, independent of data structures.

There are recent theories (Liu et al., 2025b) and experiments (Barkeshli et al., 2026) arguing similarly that neural scaling laws can emerge without power-law structures in the data, while they cannot explain the observed small time scaling exponent.

Regarding our specific prediction, $1/3$, for the scaling exponent, previous empirical papers offer various opinions. Kaplan et al. (2020) implicitly assumed zero irreducible loss, i.e., zero entropy of the target distribution, which may have led to an inaccurately fitted exponent, $0.095$. Hoffmann et al. (2022) considered the irreducible loss and obtained an exponent $0.28$, close to $1/3$. One of the first works reporting power-law scaling in language modeling (Hestness et al., 2017) showed exponents $0.30$ and $0.36$ in its first figure, both close to $1/3$. In retrospect, signatures of the $1/3$ time scaling may have already been present in the literature for nearly a decade.

## 6. Discussion

We have identified one mechanism why LLM training exhibits power-law loss scaling with time and determined the exponent. Our minimal toy model, i.e., a single layer mimicking the LM head, reproduces power-law training behaviors. Analysis reveals that **softmax and cross-entropy non-linearities** are the key: they produce power-law vanishing losses and gradients when outputting peaked distributions, inevitably leading to loss $L \sim \tau^{-1/3}$.

Our work has several limitations that open avenues for future research. We analyze gradient flow, but the effects of gradient noise from limited batch size or large learning rates warrant investigation. The solvable aligned student ansatz cannot explain why the rotation of parameters may also lead to $1/3$ time scaling as observed in the toy model.

We want the minimum example of power-law training to find the most fundamental ingredients. Yet, one of the next steps is to prove the robustness of the mechanism found theoretically, like extending the toy model to have more layers. New phenomena in the toy model may also be insightful for other aspects of training. We found large learning rate prevents rotation in the low-temperature regime. A learning rate decay may be useful not only at the edge of stability (Cohen et al., 2021; Wen et al., 2024; Liu et al., 2025c) but also under vanishing gradients to increase the signal-to-noise ratio. Beyond toy models, we hope that more LLMs at different training steps can be published for analysis.

Our theory has concrete implications for LLM training and architecture. First, training LLMs should avoid large weight decays as LLMs need to reach large logit values for peaked distributions. Decreasing learning rates or using weight decay towards the moving average (Liu et al., 2025a) or restricting hidden states on the same sphere (Loshchilov et al., 2024) may help with numeric stability. For future training, we hope to have new optimizers strong at rotating parameters. If the optimizer can outperform the gradient flow, we may have faster loss scaling. And if the parameters can converge in terms of angle first, we may stop training early and tune the temperature for accuracy manually. Finally, future architectures may make use of the token hierarchy to avoid outputting a distribution over the whole vocabulary. Then LLMs may be out of the low-temperature regime, fundamentally changing the loss scaling.

Our work emphasizes the critical roles of architectural factors, i.e., softmax and cross-entropy, in shaping training dynamics. We hope these insights will support the continued development of LLMs.

## Acknowledgements

The authors acknowledge the MIT Office of Research Computing and Data for providing high-performance computing resources that have contributed to the research results reported within this paper. The authors thank the anonymous reviewers for their helpful suggestions. The authors declare no competing interests. Y. L. thanks Qiyao (Catherine) Liang, Jinyeop Song, Mehran Kardar, Nicolas Zucchet, and Amer Al-Hiyasat for insightful discussions. J. G. thanks the Schmidt Science Polymath Award for funding. C.P. is supported by an NSF CAREER Award (IIS-2239780), DARPA grants DIAL-FP-038 and AIQ-HR00112520041, the Simons Collaboration on the Physics of Learning and Neural Computation, and the William F. Milton Fund from Harvard University. This work has been made possible in part by a gift from the Chan Zuckerberg Initiative Foundation to establish the Kempner Institute for the Study of Natural and Artificial Intelligence.

## Impact Statement

This paper presents work whose goal is to advance the field of Machine Learning. There are many potential societal consequences of our work, none of which we feel must be specifically highlighted here.

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

# A. Theoretical Analysis

## A.1. Gradient Flow

The discrete gradient descent dynamics of the student is given by

$$W_{t+1} = W_t - c_{\text{eff}} \eta_t \nabla L, \tag{21}$$

where $W_t$ is the weight at step $t$, $c_{\text{eff}}$ is introduced for generality, and $\eta_t$ is the learning rate at step $t$. When using gradient descent as our optimizer, the above equation should have $c_{\text{eff}} = 1$. For advanced optimizers like Adam, this equation does not describe the actual update rules. Yet, interestingly, as our experiments suggest, the average trajectory of Adam may still follow the gradient, making the above equation a good effective description. Intuitively, Adam still follows the gradient by zigzag, which may be due to the small gradient signal and the misalignment between the coordinate directions and the gradient direction. If the gradient descent dynamics is an effective description, the effective step size can be different from the learning rate we set for the Adam optimizer, i.e., $c_{\text{eff}} \neq 1$.

In this paper, Equation (21) is a basic assumption of the dynamics. We attempted to analyze the consequences of this assumption and then compare the theoretical results with experimental findings. It was essentially the agreement between theory and experiments that suggested the assumption might be reasonable. The mechanistic reasons why Equation (21) can describe Adam and why $c_{\text{eff}}$ is a constant independent of $t$ require future studies.

Taking the continuous limit when $\eta_t$ is small, we first define the dynamic time as $\tau(t) = \int_0^t \eta_{t'} dt' \approx \sum_{t'=0}^t \eta_{t'}$. Noticing that $\eta_t$ is $d\tau$ since $dt$ here is simply 1, we can rewrite Equation (21) as

$$\frac{dW}{d\tau} = -c_{\text{eff}} \nabla L, \tag{22}$$

where $\nabla$ is taking the gradient with respect to the student weight.

We next study the gradient. For a given input,

$$L(x) = \sum_{i=1}^n p_i(x) \ln \frac{p_i(x)}{q_i(x)}. \tag{23}$$

Note that $q_i$ ($i = 1, 2, ..., n - 1$) are independent as $q_n = 1 - \sum_{j=1}^{n-1} q_j$. We have

$$\frac{\partial L(x)}{\partial q_i} = -\frac{p_i}{q_i} + \frac{p_n}{q_n}, \; i = 1, 2, ..., n - 1. \tag{24}$$

The next step in the chain rule is

$$\frac{\partial q_i}{\partial y_j} = \delta_{ij} q_i - q_i q_j. \tag{25}$$

And finally,

$$\frac{\partial y_j}{\partial W_{km}} = \delta_{jk} x_m. \tag{26}$$

Based on the chain rule,

$$\frac{\partial L(x)}{\partial W_{km}} = \sum_{i=1}^{n-1} \sum_j \frac{\partial L(x)}{\partial q_i} \frac{\partial q_i}{\partial y_j} \frac{\partial y_j}{\partial W_{km}} = (-p_k + q_k) x_m. \tag{27}$$

We therefore have

$$\nabla L(x) = (-p + q)x^T, \; \nabla L = \langle (-p + q)x^T \rangle_x, \tag{28}$$

and

$$\frac{dW}{d\tau} = c_{\text{eff}} \langle (p - q)x^T \rangle_x. \tag{29}$$

## A.2. Aligned Student Ansatz

Recall that the aligned student ansatz means that the student's weight is proportional to the teacher's. The teacher has weight $W^* = \frac{1}{\sqrt{m}} \hat{W} \beta^*$. The student weight is then $W(t) = \frac{1}{\sqrt{m}} \hat{W} \beta(t)$, where $\beta(t)$ is the student inverse temperature, trying to approach $\beta^*$. To derive the effective dynamics of $\beta$, we substitute the ansatz form into Equation (29). We then have

$$\frac{\hat{W}}{\sqrt{m}} \frac{\mathrm{d}\beta}{\mathrm{d}\tau} = c_{\mathrm{eff}} \langle (p-q) x^T \rangle_x, \tag{30}$$

and

$$\frac{\hat{W}^T \hat{W}}{\sqrt{m}} \frac{\mathrm{d}\beta}{\mathrm{d}\tau} = c_{\mathrm{eff}} \langle \hat{W}^T (p-q) x^T \rangle_x, \tag{31}$$

and

$$\frac{\mathrm{Tr}(\hat{W}^T \hat{W})}{\sqrt{m}} \frac{\mathrm{d}\beta}{\mathrm{d}\tau} = c_{\mathrm{eff}} \langle (p-q)^T \hat{W} x \rangle_x. \tag{32}$$

To obtain the typical behavior of $\beta$, we take average over teacher randomness $\hat{W}$, and obtain

$$\frac{mn}{\sqrt{m}} \frac{\mathrm{d}\beta}{\mathrm{d}\tau} = c_{\mathrm{eff}} \langle (p-q)^T \hat{W} x \rangle_{x,\hat{W}}. \tag{33}$$

Recall our definition of energy $\epsilon_i = -\frac{1}{\sqrt{m}} (\hat{W} x)_i$, we can rewrite the above equation as

$$\frac{\mathrm{d}\beta}{\mathrm{d}\tau} = -\frac{c_{\mathrm{eff}}}{n} \langle (p-q)^T \epsilon \rangle, \tag{34}$$

where we use $\langle \cdot \rangle$ to represent the average over all sampling randomness, i.e., $\langle \cdot \rangle_{x,\hat{W}}$.

We next introduce

$$Z(\beta) = \sum_i e^{-\beta \epsilon_i}. \tag{35}$$

Note that

$$p_i = \frac{e^{-\beta^* \epsilon_i}}{Z(\beta^*)}, \; q_i = \frac{e^{-\beta \epsilon_i}}{Z(\beta)}, \tag{36}$$

one can have

$$p^T \epsilon = \sum_i p_i \epsilon_i = -\frac{\partial \ln Z}{\partial \beta}(\beta^*), \; q^T \epsilon = \sum_i q_i \epsilon_i = -\frac{\partial \ln Z}{\partial \beta}(\beta). \tag{37}$$

We therefore have

$$\frac{\mathrm{d}\beta}{\mathrm{d}\tau} = \frac{c_{\mathrm{eff}}}{n} \left( \frac{\partial \langle \ln Z \rangle}{\partial \beta}(\beta^*) - \frac{\partial \langle \ln Z \rangle}{\partial \beta}(\beta) \right). \tag{38}$$

Now, Equation (35) and Equation (38) are already closed for $\beta$, and $\langle \cdot \rangle$ is equivalent to $\langle \cdot \rangle_\epsilon$ – averaging over energies $\epsilon_i$.

For more intuitions and future convenience, we further introduce free energy $F$, defined through

$$\langle \ln Z \rangle = -\beta F(\beta). \tag{39}$$

Following the analogy from statistical physics, one can define the internal energy as

$$U = -\frac{\partial \langle \ln Z \rangle}{\partial \beta} = \frac{\partial (\beta F)}{\partial \beta}. \tag{40}$$

We can also define entropy $S$ as

$$S(\beta) = \langle -\sum_i q_i \ln q_i \rangle. \tag{41}$$

And the following relation is true:

$$F = U - \frac{1}{\beta} S. \tag{42}$$

To study the loss dynamics, we also need the typical behavior of loss as a function of $\beta$, i.e.,

$$L(\beta) = \left\langle \sum_{i=1}^{n} p_i \ln \frac{p_i}{q_i} \right\rangle = U(\beta^*)(\beta - \beta^*) + \beta^* F(\beta^*) - \beta F(\beta). \tag{43}$$

The second equation above can be directly obtained based on the definitions of $p$ and $q$. Note that then

$$\frac{\mathrm{d}L(\beta)}{\mathrm{d}\beta} = U(\beta^*) - U(\beta), \tag{44}$$

we can obtain the following from Equation (38) and the definition of $U$:

$$\frac{\mathrm{d}\beta}{\mathrm{d}\tau} = -\frac{c_{\mathrm{eff}}}{n} \frac{\mathrm{d}L(\beta)}{\mathrm{d}\beta}. \tag{45}$$

To understand the loss dynamics, we need to understand $F$ or $\langle \ln Z \rangle$, which can lead to $U$ and together with $U$ define $L$.

### A.3. Low Temperature Expansion

We are not able to solve $F$ analytically for all $\beta$, but we can consider the limit $\beta \to \infty$, the low temperature limit. By definition,

$$-\beta F(\beta) = \langle \ln \sum_i e^{-\beta \epsilon_i} \rangle. \tag{46}$$

For each $\{\epsilon_i\}_{i=1}^{n}$ or $\epsilon = [\epsilon_1, ..., \epsilon_n]^T$, we use $\epsilon^\uparrow$ for the corresponding sorted energies, such that $\min_i \epsilon_i = \epsilon_1^\uparrow \leq \epsilon_2^\uparrow \leq \cdots \leq \epsilon_n^\uparrow$. Then, we can write

$$\langle \ln \sum_i e^{-\beta \epsilon_i} \rangle = -\beta \langle \epsilon_1^\uparrow \rangle + \left\langle \ln \left(1 + \sum_{i=2}^{n} e^{-\beta \Delta \epsilon_i}\right) \right\rangle, \tag{47}$$

where $\Delta \epsilon_i = \epsilon_i^\uparrow - \epsilon_1^\uparrow$ are called energy gaps. With large $\beta$ and a finite $n$, $\sum_{i=2}^{n} e^{-\beta \Delta \epsilon_i}$ is small, such that

$$\left\langle \ln \left(1 + \sum_{i=2}^{n} e^{-\beta \Delta \epsilon_i}\right) \right\rangle \approx \sum_{i=2}^{n} \langle e^{-\beta \Delta \epsilon_i} \rangle. \tag{48}$$

The key is then to understand

$$\langle e^{-\beta \Delta \epsilon_i} \rangle = \int_0^\infty e^{-\beta \Delta \epsilon_i} \rho_{\epsilon_i}(\epsilon_i) \mathrm{d}\epsilon_i, \tag{49}$$

where $\rho_{\epsilon_i}(\cdot)$ is the probability density function of $\epsilon_i$.

We first consider $\epsilon_2$ since it is smallest and $e^{-\beta \Delta \epsilon_2}$ dominates $\sum_{i=2}^{n} e^{-\beta \Delta \epsilon_i}$. The mild assumption we are going to use is that $\rho_{\epsilon_2}(0) > 0$ and $\rho_{\epsilon_2}$ is continuous around 0. In our toy model setup, where $\epsilon_i$ are i.i.d. Gaussian, this assumption is satisfied. For LLMs, this is hard to verify via experiments. Yet, conceptually, the assumption means that among the contexts, the most likely next token and the second most likely next token have non-zero probability to be equally likely. Heuristically, we can imagine a context "my favorite pet is a" whose next possible word may be equally likely to be "dog" or "cat". Therefore, we think the assumption is relevant and generic, and will proceed with it, yielding

$$\int_0^\infty e^{-\beta \Delta \epsilon_2} \rho_{\epsilon_2}(\epsilon_i) \mathrm{d}\epsilon_2 \approx \int_0^{\beta^{-1}} \rho_{\epsilon_2}(\epsilon_2) \mathrm{d}\epsilon_2 \approx \rho_{\epsilon_2}(0) \beta^{-1}, \tag{50}$$

where the first approximation is due to the property of the exponential function, and the second is due to large $\beta$. For other $\Delta \epsilon_i$ integral, they can also yield a $\beta^{-1}$ term as leading term or a much smaller term if $\rho_{\epsilon_i}(0) = 0$. Regardless, we can now conclude that

$$\sum_{i=2}^{n} \langle e^{-\beta \Delta \epsilon_i} \rangle \propto \beta^{-1} \tag{51}$$

is a good approximation when $\beta$ is large. Finally, we can have

$$-\beta F(\beta) = -\beta \langle \epsilon_1^\uparrow \rangle + c_2 \beta^{-1} + \cdots, \tag{52}$$

where $c_2$ is some positive constant and $\cdots$ contains high-order terms. The free energy is then

$$F(\beta) = -c_0 - c_1\beta^{-1} - c_2\beta^{-2} + \cdots, \tag{53}$$

with $c_0 = -\langle \epsilon_1^\uparrow \rangle$ and $c_1 = 0$. The above derivation assumed no degeneracy for simplicity. If we allow $\rho_{\Delta\epsilon_i}$ to have a delta function at zero, while defining $\rho_{\Delta\epsilon_i}(0)$ as the continuous part of the density function at zero, we will have $c_1 \neq 0$, which is more general, and all other arguments unchanged. We therefore complete the proof of the low-temperature expansion argument raised in Section 3.

After the derivation, we summarize and discuss

---

### Notes on the low-temperature expansion

- **Conditions**.—The expansion above depends on the property of exponential functions (i.e., non-linearity) and generic data properties. In particular, the first data property is that the output distribution is peaked (i.e., low temperature). The second data property is that there is a non-zero density in the data, such that the most likely output choice and the second most likely one are equally likely. The second property leads to the non-zero $c_2$ in free energy expansion, which is important for the loss scaling later. Heuristically, the second property requires the dataset to be diverse enough to include vague cases, which seems to be generic for large natural language datasets.

- **Analyticity**.—In Equation (53), terms in "$\cdots$" are higher order terms, decaying faster than $\beta^{-2}$. If we only assume that density functions $\rho_{\Delta\epsilon_i}$ are continuous around $0$, the higher order terms are not necessarily a polynomial of $\beta^{-1}$. If we further assume that $\rho_{\Delta\epsilon_i}$ are analytic around $0$, $F(\beta)$ can be approximated by an analytic function of $\beta^{-1}$, and the higher order terms will be $\beta^{-3}$, $\beta^{-4}$, etc.

- **Universality**.—In statistical physics, universality means that critical exponents depend only on generic properties of a system, not its microscopic details; many universality classes can exist, each with its own exponent shared across a wide range of systems. The situation here is analogous: whenever the generic conditions above are satisfied, the term $c_2\beta^{-2}$ in Equation (53) persists, giving rise to the $1/3$ time scaling in loss. Microscopic details of a dataset may affect $c_2$ but not the exponent, so all such datasets belong to the same universality class. Different universality classes arise when $c_2 = 0$, and if analyticity is assumed, in which case the leading power law comes from a higher-order term and the loss scaling exponent changes. We have no reason to assume $c_2 = 0$ here, but understanding boundaries of universality classes and how real datasets map onto these classes are important directions for future work.

---

Given the free energy expansion, following the definition of internal energy, we can easily arrive

$$U(\beta) = -c_0 + c_2\beta^{-2} + \cdots. \tag{54}$$

When both student and teacher are in the low-temperature regime (we can apply the above expansion to both $\beta$ and $\beta^*$), we substitute the low-temperature expansions of $F(\beta)$, $F(\beta^*)$, $U(\beta)$, and $U(\beta^*)$ to loss and have

$$L = c_2(\beta^{-1} - \beta^{*-1}) + c_2(\beta - \beta^*)\beta^{*-2} + \cdots, \tag{55}$$

and then the negative gradient

$$-\frac{\mathrm{d}L}{\mathrm{d}\beta} = U(\beta) - U(\beta^*) = c_2(\beta^{-2} - \beta^{*-2}) + \cdots. \tag{56}$$

In the **intermediate regime** where $\beta$ is large enough while much smaller than $\beta^*$, we further have

$$L \approx c_2\beta^{-1}, \quad -\frac{\mathrm{d}L}{\mathrm{d}\beta} \approx c_2\beta^{-2}, \tag{57}$$

subsequently,

$$\beta \sim \tau^{1/3}, \tag{58}$$

and finally

$$L \sim \beta^{-1} \sim \tau^{-1/3}. \tag{59}$$

### A.4. High Temperature Expansion

In the limit of high temperature, $\beta \to 0$, we can expand $\ln Z$ directly with $\beta$.

$$
\begin{aligned}
\ln Z &= \ln \sum_i e^{-\beta \epsilon_i} \\
&= \ln n + \ln \left( 1 + \frac{\sum_i -\beta \epsilon_i + \frac{1}{2}\beta^2 \epsilon_i^2 + \cdots}{n} \right) \\
&= \ln n - \frac{1}{n}\sum_i \beta \epsilon_i + \frac{1}{2n}\sum_i \beta^2 \epsilon_i^2 - \frac{1}{2n^2}(\sum_i -\beta \epsilon_i + \cdots)^2 + \cdots .
\end{aligned}
\tag{60}
$$

Taking the average over $\epsilon$, the leading term is $O(\beta^2)$,

$$
\begin{aligned}
\langle \ln Z \rangle &= \ln n + \frac{1}{2}\beta^2 - \frac{1}{2n}\beta^2 + \cdots \\
&\approx \ln n + \frac{1}{2}\beta^2.
\end{aligned}
\tag{61}
$$

The $\approx$ is true for large $n$ and small $\beta$. We then have

$$
F = -\frac{1}{\beta}\ln n - \frac{1}{2}\beta, \ U(\beta) = -\beta.
\tag{62}
$$

The loss is then

$$
L(\beta) = U(\beta^*)(\beta - \beta^*) + \beta^* F(\beta^*) + \ln n + \frac{1}{2}\beta^2,
\tag{63}
$$

with gradient

$$
-\frac{\mathrm{d}L}{\mathrm{d}\beta} = -U(\beta^*) - \beta.
\tag{64}
$$

When $\beta^*$ is large while $\beta$ is very small (e.g., in learning peaked distributions, with small initialization, and at early time), we use $-c_0$ to approximate $U(\beta^*)$ and $F(\beta^*)$ and have

$$
L = \ln n - c_0\beta + \frac{1}{2}\beta^2 \approx \ln n - c_0\beta,
\tag{65}
$$

$$
-\frac{\mathrm{d}L}{\mathrm{d}\beta} = c_0 - \beta.
\tag{66}
$$

In such a case, $\beta$ will initially grow linearly in $\tau$, and loss $L$ will decrease linearly with $\tau$, where there is no power law.

If both $\beta^*$ and $\beta$ are small (e.g., in learning flat distributions), we will have

$$
L = \frac{1}{2}(\beta - \beta^*)^2,
\tag{67}
$$

and

$$
-\frac{\mathrm{d}L}{\mathrm{d}\beta} = \beta^* - \beta.
\tag{68}
$$

We can then have the solution

$$
\beta^* - \beta(t) = (\beta^* - \beta_0)e^{-\frac{c_{\mathrm{eff}}}{n}\tau},
\tag{69}
$$

and

$$
L = \frac{1}{2}(\beta^* - \beta_0)^2 e^{-\frac{2c_{\mathrm{eff}}}{n}\tau}.
\tag{70}
$$

That is, in the high-temperature regime, with our setup, the convergence is exponential in time.

## B. Toy Model Analysis

### B.1. Numerical Calculations

To numerically calculate loss value $L(\beta)$ and gradient $\frac{\mathrm{d}L}{\mathrm{d}\beta}$ as a function of $\beta$ under the aligned student ansatz, we fix $m = 32$ and $n = 128$ as is in our training experiments. For each given $\beta^*$, we choose 100 points linearly from 0 to $\beta^*$. Then, at each $\beta$ grid point, we sample 10 different $\hat{W}$. For each $\hat{W}$, we sample 1024 different inputs $x$ to calculate the loss as the KL-divergence. Once the logits are calculated, we can calculate the gradient based on the right-hand side of Equation (34). Finally, we take the mean loss and mean gradient value that only depend on $\beta$ and $\beta^*$. These calculations gave Figure 3, and we used `polyfit` in log-log scale to get the exponents. The code is in `test-2.ipynb`.

Similarly, based on the relation $U(\beta) = \langle q^T \epsilon \rangle$, we can also obtain the numerical values of $U(\beta)$. One can see that $U(\beta) \approx -\beta$ is true for small $\beta$ (Section D.1), supporting the theory of high-temperature expansion.

### B.2. Optimization Experiments

The toy model has a simple architecture, i.e., one linear layer followed by the softmax non-linearity, which can be easily implemented with PyTorch (details in code). Here, we provide an overview of important hyperparameters to tune and the quantities to save.

We fixed the hidden dimension $m = 32$ and $n = 128$. The numbers are not special and are kept small for efficiency.

We tested both SGD and Adam. For SGD, a range of learning rates from 1e-2 to 1e+1 are tested. For Adam, learning rates from 1e-3 to 1 are tested.

For both teacher and student, we initialized the weights based on the default initialization of PyTorch, i.e., i.i.d from $\mathcal{U}(-1/\sqrt{m}, 1\sqrt{m})$, where each element has standard deviation $1/\sqrt{3m}$. We can then multiply a number by the student weight to change the initialization scale. We divide the teacher weight by a variable called `temperature`. The inverse temperature, as standard deviation of logits, in this case, is actually $\beta^* = \frac{1}{\sqrt{3}*\text{temperature}}$. Although we did not faithfully follow the theory to sample the weights as Gaussian. It does not matter in the end because our derivation of the theory only used the i.i.d. property, and the logits will be Gaussian due to the central limit theorem. This is also called universality – details of the distribution do not matter, and we just need to care about the moments. The `temperature` was scanned from 1e-3 to 1.

The batch sizes are tested from 128 to 2048, which yield no big differences. We primarily use a batch size of 1024 to ensure small noise similar to LLM training while making the training efficient.

For different cases, the number of total training steps ranges from 1000 to 100000.

We evaluated the loss, logit standard deviation, and student weight norm during the training, which were saved for further analysis.

## C. LLM Experiments

We analyzed, from the smallest to the largest, `EleutherAI/pythia-70m`, `410m`, `2.8b`, and `12b` (Biderman et al., 2023), and additionally `allenai/OLMo-2-1B`, `7B`, and `13B` (OLMo et al., 2024). For each model size, the script evaluates a sequence of training checkpoints. The code first defines a list of checkpoint identifiers (we try to make the chosen steps evenly on the log scale) using the Pythia convention `stepX`, where each element of `steps` corresponds to a distinct model revision hosted on the Hugging Face Hub.

Text is drawn from the `HuggingFaceFW/fineweb` dataset using streaming mode. Streaming avoids materializing the dataset on disk and provides an iterator that yields samples sequentially. For each model revision, the iterator is re-initialized so that evaluation always begins from the start of the stream. The evaluation workload is defined in terms of documents and batches: a batch contains `batch_size` documents, and the script processes `num_docs` documents in total (i.e., `num_docs` / `batch_size` batches). Each document is taken from the `text` field of the dataset sample. We evaluated the same 2M tokens for different models.

Each batch of raw strings is tokenized with the model's tokenizer using fixed-length padding and truncation to `max_length`. This produces `input_ids` and an `attention_mask` of shape (`batch_size`, `seq_len`). Because padding is

introduced to reach a uniform length, the script ensures the tokenizer has a padding token; if none is defined, it reuses the end-of-sequence token as `pad_token`. The labels for next-token prediction are created by cloning `input_ids` and then replacing padded positions with `-100`, which is the standard ignore index used by Hugging Face loss functions so that padded tokens do not contribute to the cross-entropy objective.

To analyze logits only where next-token prediction is well-defined on non-padding tokens, the script constructs a `valid_mask` over time steps. This mask selects positions where both the current token and the next token are real (non-padding) tokens, implemented by requiring that `attention_mask` is positive at positions $t$ and $t+1$. The model is then run under `torch.no_grad()` in evaluation mode, passing both inputs and labels. This produces the scalar loss (`out.loss`) and the full logits tensor (`out.logits`) with shape (`batch_size, seq_len, vocab_size`).

For statistics, the code aligns logits and labels for next-token prediction by dropping the final time step of logits and shifting labels forward by one position. It then applies `valid_mask` so that the analysis is performed on a flattened set of valid token positions across the batch, yielding a matrix `logits` of shape (`num_valid_tokens, vocab_size`). These logits are cast to `float32` for numerically stable moment calculations. The corresponding ground-truth next tokens are extracted as `valid_labels`, and the logit assigned to the correct next token is gathered into `correct_logits`. The script accumulates the total count of valid token positions so that all reported quantities can be normalized per token at the end.

Several per-token distribution summaries are computed from the vocabulary logits. The code accumulates the standard deviation across the vocabulary dimension, `logits.std(dim=-1)`, as a measure of how peaked or spread out the predictive distribution is in logit space, as well as the vocabulary mean `logits.mean(dim=-1)`. In addition to these moments, it tracks quantities that compare extremes or the correct token to the typical logit level. The variable named `skewness` is computed as `correct_logits - mean_logit` (summed over valid tokens), which captures how far above (or below) the average vocabulary logit the model places the true next token. The variable `skewness_extreme` is computed as `max_logit - mean_logit`, which measures how dominant the single most-preferred token is relative to the average. Finally, `logit_range` records `max_logit - min_logit` across the vocabulary as a simple dynamic-range statistic. The cross-entropy loss is accumulated in a token-weighted way by multiplying `out.loss` by the number of valid token positions so that averaging at the end reflects the same weighting as the other per-token statistics.

Because tokenization uses fixed-length padding, the script divides several intermediate sums by `max_length` within the batch loop, and then multiplies by `max_length` when forming the final averages. This bookkeeping ensures that the final reported values correspond to averages per token position rather than being inadvertently scaled by the fixed sequence length. After all batches for a revision are processed, the model object is deleted to release GPU memory, and the wall-clock time for processing `num_docs` documents is printed as a simple throughput diagnostic.

At the end of the full sweep over revisions, the script normalizes each accumulated statistic by `total_tokens` to obtain per-token averages for every checkpoint. It then saves a single PyTorch checkpoint file containing the resulting vectors over training steps, including the averaged loss, logit standard deviation, logit mean, total token counts, the correct-token margin relative to the mean (`skewness`), the maximum-token margin relative to the mean (`skewness_extreme`), and the vocabulary logit range. This file serves as the compact artifact used for downstream plotting and analysis of how the predictive distribution evolves over training.

# D. Data Analysis and Supplementary Results

## D.1. Adam Scanning Temperatures and Learning Rates

The framework of experiments is in Section B.2. We specify here that we used the Adam optimizer, and scanned $\beta^*$ from $\sim 0.58$ to $\sim 577$, and learning rates from 1e-3 to 1. There are 8 different $\beta^*$ evenly spaced in log scale and 12 different learning rates evenly spaced in log scale, such that there are $12 \times 8 = 96$ cases. The number of training steps is 2000, and the batch size is 1024 for all cases. We initialize all student weight at zero. The learning rate $\eta$ does not vary during training, and the dynamic time $\tau = t\eta$ at the training step $t$. The code is in `exp-1.py`.

We plot the test loss as a function of $\tau$ in Figure 7, and $\beta$ as a function of $\tau$ in Figure 8. From left to right, temperature $1/\beta^*$ increases. In each panel, we use color to distinguish learning rates. The values in the color bar represent $\log_{10} \eta$. The dashed line represents a power-law decay with exponent $1/3$ in Figure 7 and a power-law growth with exponent $1/3$ in Figure 8. One can see that as long as the learning rates are small, the curves collapse. And when the temperature is low (left), both

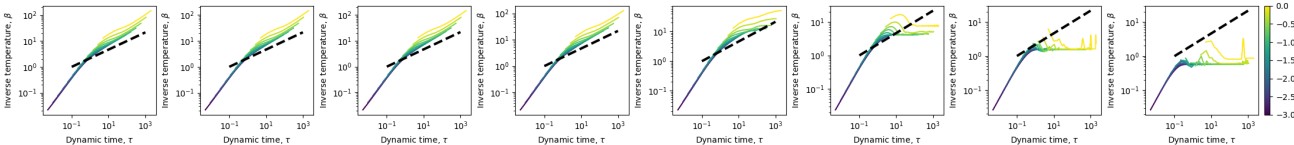

*Figure 7.* Loss as a function of $\tau$. From left to right, temperature $1/\beta^*$ increases. In each panel, we use color to distinguish learning rates. The values in the color bar represent $\log_{10}\eta$. The dashed line represents a power law with exponent $1/3$. One can see that as long as the learning rates are small, the curves collapse. And when the temperature is low (left), both loss and $\beta$ agree with our predictions from the aligned student ansatz for the collapsed lines.

*Figure 8.* Student inverse temperature as a function of $\tau$. From left to right, temperature $1/\beta^*$ increases. In each panel, we use color to distinguish learning rates. The values in the color bar represent $\log_{10}\eta$. The dashed line represents a power law with exponent $1/3$. One can see that as long as the learning rates are small, the curves collapse. And when the temperature is low (left), both loss and $\beta$ agree with our predictions from the aligned student ansatz for the collapsed lines.

loss and $\beta$ agree with our predictions from the aligned student ansatz for the collapsed lines. We therefore conclude that as long as the learning rate is low, the dynamics effectively follow the gradient flow. For large learning rates, the noise is too strong, and the weights have difficulty following the gradient to align or rotate, leading to a slowly decreasing loss and non-collapsing loss curves. The optimal learning rate, i.e., the learning rate leading to the lowest loss given the same step, from Figure 7, seems to be the largest learning rate that can still follow gradient flow (or collapse with curves with smaller learning rates).

We picked $\eta = 0.023$, which is sufficiently small for most temperatures, and all 8 temperatures from Figure 7 to show in Figure 1.

We picked $\eta = 0.001, 0.0066, 0.081, 0.53$ and largest $\beta^* = 577.3$ from Figure 7 and Figure 8 to show in Figure 4. We fitted the exponent from the curve with the optimal learning rate, as it is deepest in the low temperature regime, given the training steps.

In the main text, we studied the low-temperature expansion. Here, we also analyze the high-temperature results. First, we numerically calculate the internal energy $U$ as described in Section B.1. Our theory predicts $U(\beta) = -\beta$ for small $\beta$ (Section A.4), which agrees with the numerical values (Figure 9).

We next check the training results in the high-temperature regime. We pick the two $\beta^*$ that are smaller than $\sqrt{2\ln n}$ from Figure 8 and the small learning rates to show in Figure 10. Too large a learning rate will lead to non-smooth dynamics. We predict that $\beta^* - \beta$ will converge to zero with a rate $c_{\text{eff}}/n$ in $\tau$ (Section A.4). For each $\beta^*$, once we plot with $\tau$, we can see that in a log-linear plot that $(\beta^* - \beta)/\beta^*$ obtained from different learning rates collapse to the same line initially, supporting our theory. Interestingly, the slopes of the lines are different for different $\beta^*$, suggesting $c_{\text{eff}}$ can depend on the landscape or

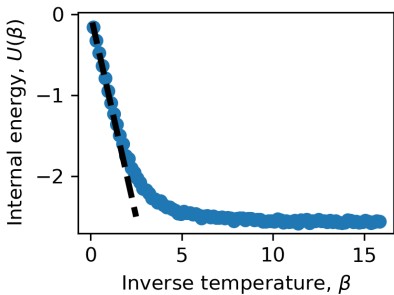

*Figure 9.* The numerical value (method in Section B.1) of internal energy supports our theory in the high-temperature regime. The dashed line represents $U(\beta) = -\beta$.

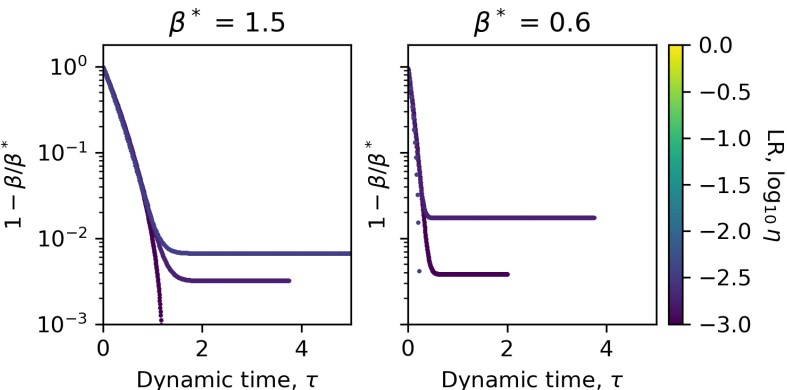

*Figure 10.* In the high temperature regime, inverse temperature $\beta$ converges exponentially to the target $\beta^*$, agreeing with the theory.

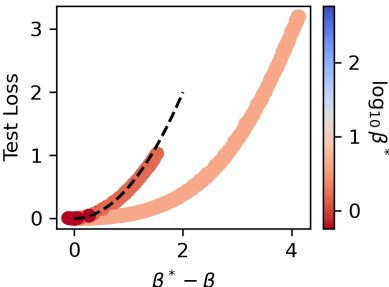

*Figure 11.* In the high temperature regime (small $\beta^*$), $L = (\beta^* - \beta)^2/2$. Data points are from Figure 7 and Figure 8. The dashed line is $L = (\beta^* - \beta)^2/2$.

$\beta^*$. We also tested the predicted relationship between $\beta^* - \beta$ and loss, $L = (\beta^* - \beta)^2/2$, and the empirical results agree well with this theory once $\beta^*$ is small enough (Figure 11). We conclude that our theory in the high-temperature regime under the aligned student ansatz can well describe the real training results.

As a side note, in the high temperature regime, the loss is basically a quadratic function, and therefore leads to exponential convergence without a power-law structure in data. Previous theories built on MSE require power-law structures in data to obtain a power-law convergence.

In the code `exp-0.py`, we use the Adam optimizer, and scanned $\beta^*$ from 9.15 to 57.7, and learning rate 3e-3. There are 8 different $\beta^*$ evenly spaced in log scale. The number of training steps is 50000, and the batch size is 1024. We saved the student weight every 200 steps. To visualize the teacher and student weights, we sample a random matrix $\Pi \in \mathbb{R}^{m \times 2}$ to project the weight $W \in \mathbb{R}^{n \times m}$ into $W\Pi \in \mathbb{R}^{n \times 2}$. And we can plot each projected row in a two-dimensional plot. The results are in Figure 12. One can see that the teacher and the student weights are quite aligned regardless of the temperature. We chose $\beta^* = 34.1$ to show in Figure 2.

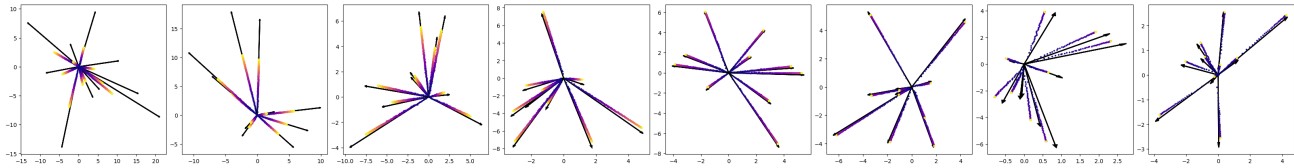

*Figure 12.* Actual training inspires the aligned student ansatz (student weight is proportional to the teacher's). From right to left, $\beta^*$ are from 9.15 to 57.7, evenly in log scale. We project rows of teacher weight $W_i^* \in \mathbb{R}^m$ and the corresponding student weight rows $W_i$ to a 2-dimensional space. The student weight is initialized at zero. The arrows are fixed teacher weight rows. The colored dots are student rows at different steps. Color bar is the same as Figure 2.

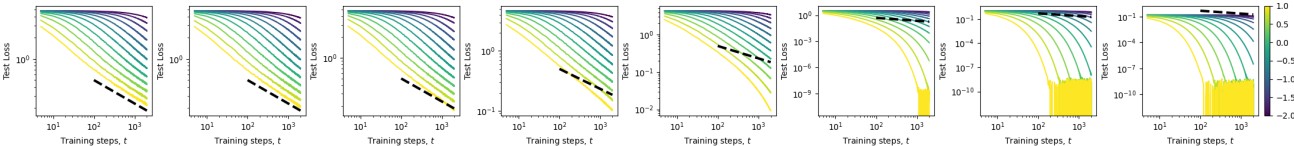

*Figure 13.* Loss as a function of step $t$. From left to right, temperature $1/\beta^*$ increases. In each panel, we use color to distinguish learning rates. The values in the color bar represent $\log_{10} \eta$. The dashed line represents a power law with exponent $1/3$. When the temperature is low (left), both loss and $\beta$ agree with our predictions.

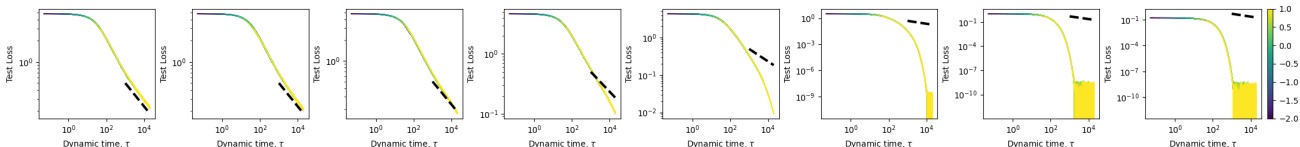

*Figure 14.* Loss as a function of $\tau$. From left to right, temperature $1/\beta^*$ increases. In each panel, we use color to distinguish learning rates. The values in the color bar represent $\log_{10} \eta$. The dashed line represents a power law with exponent $1/3$. All the curves collapse. And when the temperature is low (left), both loss and $\beta$ agree with our predictions from the aligned student ansatz for the collapsed lines.

### D.2. SGD Scanning Temperatures and Learning Rates

We did the same experiments again as Section D.1, except changing Adam to SGD and changing the learning rate range from `torch.logspace(-3, 0, steps=12)` to `torch.logspace(-2, 1, steps=12)`. The code is in `exp-1-1.py`. The loss as a function of training steps is in Figure 13. The loss as a function of dynamic time $\tau$ is in Figure 14. The inverse temperature $\beta$ as a function of training steps is in Figure 15. The inverse temperature $\beta$ as a function of dynamic time $\tau$ is in Figure 16. Overall, the SGD results agree with the theory very well, and all the curves can collapse once plotted with $\tau$ for the range of learning rates we scanned. This is not surprising, as our theory of gradient flow should well describe SGD.

We next check the training results in the high-temperature regime. We pick the two $\beta^*$ that are smaller than $\sqrt{2 \ln n}$ from Figure 16 and the small learning rates to show in Figure 17. We predict that $\beta^* - \beta$ will converge to zero with a rate $c_{\text{eff}} = 1$ for SGD in normalized time $\tau/n$ (Section A.4). For each $\beta^*$, once we plot with $\tau$, we can see that in a log-linear plot that $(\beta^* - \beta)/\beta^*$ obtained from different learning rates collapse to the same line, well supporting our theory. And the slope of the line, which should be $c_{\text{eff}} = 1$ here for SGD, is also verified. We also tested the predicted relationship between $\beta^* - \beta$ and loss, $L = (\beta^* - \beta)^2/2$, and the empirical results agree well with this theory once $\beta^*$ is small enough (Figure 18). We conclude that our theory in the high-temperature regime under the aligned student ansatz can well describe the real training results of SGD.

### D.3. Adam Scanning Initialization Scales

The framework of experiments is in Section B.2. We specify here that we used the Adam optimizer, and fixed $\beta^*$ as 96.2, scanned initialization ratio $\beta_0/\beta^*$ from 0 to 1, scanned learning rates from 1e-3 to 1. There are 8 different $\beta_0/\beta^*$ evenly spaced in linear scale and 12 different learning rates evenly spaced in log scale, such that there are $12 \times 8 = 96$ cases. The number of training steps is 100000, and the batch size is 1024 for all cases. We initialize all student weight at zero. The learning rate $\eta$ does not vary during training, and the dynamic time $\tau = t\eta$ at the training step $t$. The code is in `exp-2.py`.

We plot the test loss as a function of $\tau$ in Figure 19, and $\beta$ as a function of $\tau$ in Figure 20. From left to right, the initialization

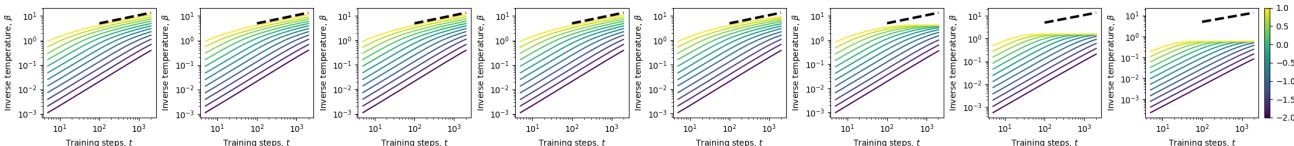

*Figure 15.* Inverse temperature $\beta$ as a function of step $t$. From left to right, temperature $1/\beta^*$ increases. In each panel, we use color to distinguish learning rates. The values in the color bar represent $\log_{10} \eta$. The dashed line represents a power-law growth with exponent $1/3$. When the temperature is low (left), both loss and $\beta$ agree with our predictions.

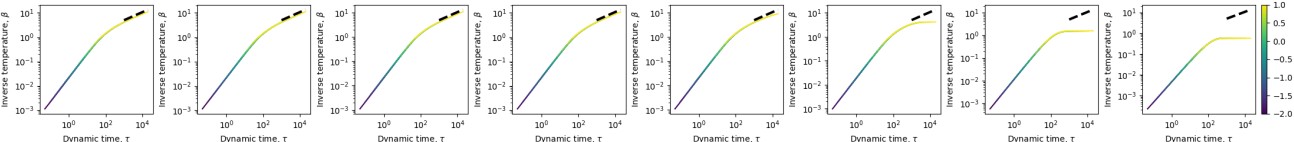

*Figure 16.* Inverse temperature $\beta$ as a function of $\tau$. From left to right, temperature $1/\beta^*$ increases. In each panel, we use color to distinguish learning rates. The values in the color bar represent $\log_{10} \eta$. The dashed line represents a power-law growth with exponent $1/3$. When the temperature is low (left), both loss and $\beta$ agree with our predictions.

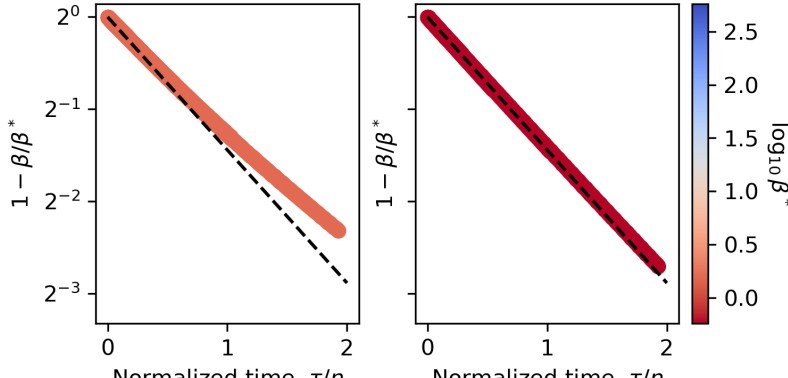

*Figure 17.* In the high temperature regime, inverse temperature $\beta$ converges exponentially to the target $\beta^*$, agreeing with the theory of SGD. The dashed line refers to the theoretical line with an exponential decay rate of $1$.

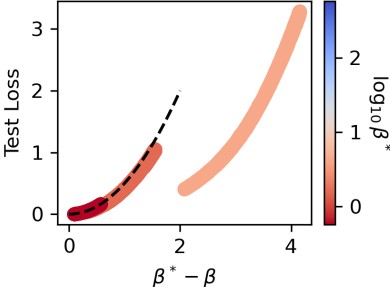

*Figure 18.* In the high temperature regime (small $\beta^*$), $L = (\beta^* - \beta)^2/2$. Data points are from Figure 14 and Figure 16. The dashed line is $L = (\beta^* - \beta)^2/2$.

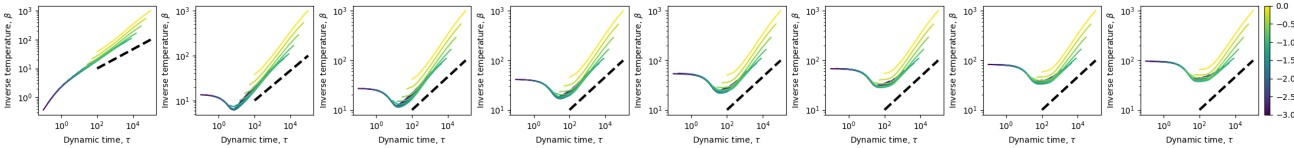

*Figure 19.* Loss as a function of $\tau$. From left to right, initialization ratio $\beta_0/\beta^*$ increases from 0 to 1. In each panel, we use color to distinguish learning rates. The values in the color bar represent $\log_{10} \eta$. The dashed line represents a power law with exponent $1/3$. One can see that as long as the learning rates are small, and if $\tau$ is large enough, the predictions from the aligned student ansatz can be correct even if they are not correct initially.

*Figure 20.* Inverse temperature $\beta$ as a function of $\tau$. From left to right, initialization ratio $\beta_0/\beta^*$ increases from 0 to 1. In each panel, we use color to distinguish learning rates. The values in the color bar represent $\log_{10} \eta$. The dashed line represents a power law with exponent $1/3$. One can see that as long as the learning rates are small, and if $\tau$ is large enough, the predictions from the aligned student ansatz can be correct even if they are not correct initially.

ratio $\beta_0/\beta^*$ increases. In each panel, we use color to distinguish learning rates. The values in the color bar represent $\log_{10} \eta$. The dashed line represents a power-law decay with exponent $1/3$ in Figure 19 and a power-law growth with exponent $1/3$ in Figure 20. One can see that as long as the learning rates are small, and if $\tau$ is large enough, the predictions from the aligned student ansatz can be correct even if they are not correct initially.

We picked $\eta = 0.081$, which is sufficiently small, and all 8 initialization ratios from Figure 19 and Figure 20 to show in Figure 5.

### D.4. Adam with Learning Rate Schedule

We repeated the same experiments in Section D.3, only adding a learning rate scheduler: for the first 1% of steps, the learning rate linearly increases from 0 to the maximum learning rate, and then follows standard cosine decay to 0.1 maximum learning rate at the end. Since a fixed learning rate of 0.081 is small enough, as shown in Section D.3 to follow gradient flow, we here fix our maximum learning rate as 0.15 such that the dynamic time $\tau$ at the end is similar. The code is in `exp-2-1.py`.

Our theory essentially predicts the dynamics as a function of the dynamic $\tau$. When we use a constant learning rate in training, $\tau$ is proportional to $t$, and we do not need to distinguish the two for the study of power laws (we focus on the exponent). However, when a step-dependent learning rate is used, $\tau$ depends on $t$ in a non-linear way, and the power laws should theoretically only exist when plotting with respect to $\tau$. The loss as a function of $\tau$ at different initialization ratios is plotted in Figure 21. And inverse temperature $\beta$ as a function of $\tau$ at different initialization ratios is plotted in Figure 22. We observe a clear power-law decay of loss with exponent $1/3$ and a power-law growth of $\beta$ with $1/3$ as in Figure 5, a and b. This suggests that our theory is indeed valid in practice even when the learning rate depends on the step.

One interesting difference is that at the very end, the losses in Figure 21 have a sharp drop, while those in Figure 5, a and b, do not. This is because at the end, $\beta \approx 100 \approx \beta^*$, and the student is already very close to the teacher and no longer in the intermediate regime, and therefore deviates from the power-law behavior. The learning rate decay makes the losses in Figure 21 able to follow the gradient flow and exhibit fast convergence. However, the learning rate in Figure 5, a and b, is too large at the end for rotation, and the dynamics already deviate from the gradient flow or are dominated by noise. Indeed, if we take a closer look at Figure 5, a and b, we will see the losses seem to converge to some non-zero value at the end after the $1/3$ power-law region, which should be due to the noise.

### D.5. Adam with Weight Decay

In adding the weight decay, we should not make it too large, otherwise the loss will saturate and no longer decrease very early. We also avoid very small weight decay, which would make training equivalent to the aligned student ansatz. Given

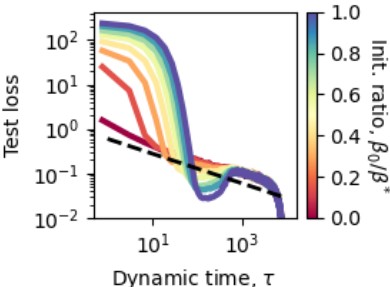

*Figure 21.* The $1/3$ time scaling can be true beyond using a constant learning rate. Initializing the student at different scales, the aligned student ansatz is wrong at first, but can be correct later, yielding the $1/3$ time scaling of loss decay with dynamic time $\tau$.

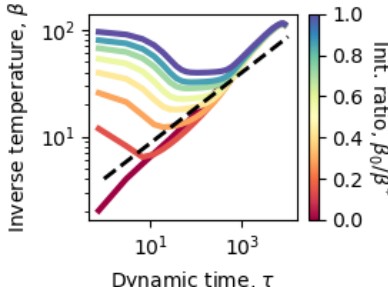

*Figure 22.* The $1/3$ time scaling can be true beyond using a constant learning rate. Initializing the student at different scales, the aligned student ansatz is wrong at first, but can be correct later, yielding the $1/3$ time scaling of $\beta$ growth with dynamic time $\tau$.

decoupled weight decay $\omega$, the range of one parameter is about $O(1/\omega)$ if the optimum is outside this range. We know the optimum, the teacher, has one parameter roughly with a norm $\beta^*/\sqrt{m}$. So, if we want the optimum to be within the range that the dynamics can reach after adding weight decay, we want $O(1/\omega) > \beta^*/\sqrt{m}$. In practice, we found that setting $\beta^* \approx 1/\omega$ is a good choice, where $\omega$ is not too large or too small. We next scanned some $\beta^*$ values. For a relatively small $\beta^* = 11.5$ and a small number of training steps 1000, increasing norm is less dominating, and the student may also have enough time to align, we can observe a loss decrease that is mainly due to rotation. The code is in `exp-3-2.py`. More specifically, we fixed $\beta^* = 11.5$, weight decay $\omega = 0.05$, a batch size 2048, and a number of training steps 1000. We scanned learning rates in `torch.logspace(-3, 0, steps=13)`. For clarity, we plotted results from learning rates $0.001, 0.006, 0.1, 0.56$ in Figure 5, c and d. The exponent fitting is done by `np.polyfit` in the log-log plot, and the dashed lines cover the region of fitting.

### D.6. Toy Model with Power Laws in Data

In previous theory and experiments on the toy model, we have shown that the emergence of the $1/3$ loss scaling does not require the power laws assumed in the data. We next test whether power laws in data can change the scaling exponent.

We use the same toy model framework as described in Section B.2, except changing the sampling of teacher weight $W^*$ to add power-law covariance. We first sample a random matrix with i.i.d. Gaussian elements with the shape $n \times m$. The random matrix has a singular value decomposition as $USV^T$. From this step, we can obtain two random rotation matrices $U$ and $V$. We then replace the singular values with a power law

$$S_{ii} \propto \frac{1}{i^{\alpha_*/2}}. \tag{71}$$

We rescale the new $S$ such that it has the same norm as the old one. At the end, we define $W^* \leftarrow USV^T$ with the new $S$. Following this sampling, the covariance between logits

$$\langle y^* y^{*T} \rangle_x = W^* W^{*T}, \tag{72}$$

which has power-law eigenvalues and the exponent is $\alpha_*$. Varying the data power-law exponent $\alpha_*$, we can obtain a series of learning curves (Figure 23). In these experiments, we fix $\beta^* = 577$, and a constant learning rate $0.08$. The target distribution

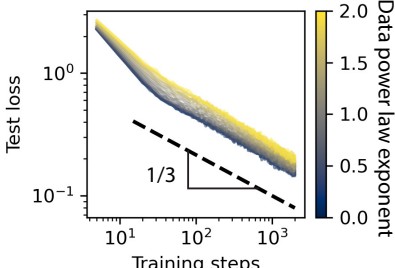

*Figure 23.* Adding a power-law structure to the data does not change the $1/3$ time scaling in loss. Data power law exponent is $\alpha_*$, which is the exponent of non-zero eigenvalues of $W^* W^{*T}$ as a function of rank.

is in the low-temperature regime and $\tau \propto t$, such that a power law fitting with training step $t$ will yield the same exponent as that with $\tau$. Scanning $\alpha_*$ in the range $0 \sim 2$, we find the loss always has $1/3$ time scaling (Figure 23). We can at least conclude that the $1/3$ time scaling is robust to a range of power-law structures in data.

We next try to analyze the robustness of the $1/3$ time scaling theoretically. By adding the power-law covariance to the data, certain directions are more important than others, and some output classes may be more frequent than others. However, this does not change the fact that the output distribution is peaked. The power-law covariance changes the boundaries where two output classes are equally likely, but does not create a probability density vacuum in space. So, the property that the energy gap has a non-zero probability density at zero does not change. Therefore, based on our theory, the power-law coefficients may be altered, but the exponent should remain $1/3$. Combined with the empirical evidence, we conclude that the $1/3$ time scaling should persist regardless of power laws in the teacher weight.

There are other ways of adding power-law structures to data. For example, we can sample $x$ from a multi-dimensional Gaussian with power-law covariance. Again, this will make the distribution of hidden states non-isotropic, but will not create a vacuum where probability density is zero. And for the logit gaps or energy gaps, the probability density at zero can change, but will not be exactly zero. Then, the $1/3$ time scaling should still be true.

### D.7. Generalized Toy Model with Residual Layers

We also tested a more generalized architecture other than the one-layer toy model in the main text. The generalized toy model has some residual layers before the head (Liu et al., 2026). We still use a teacher-student setup. The student takes input $x \in \mathbb{R}^m$ that are sampled as i.i.d. Gaussian.

$$h_0 = \mathrm{RMSNorm}(x), \tag{73}$$

$$h_l = \mathrm{MLP}_l(h_{l-1}) + h_{l-1}, \; l = 1, ..., \ell, \tag{74}$$

$$y = W \, \mathrm{RMSNorm}(h_\ell) \in \mathbb{R}^n. \tag{75}$$

Again, we call $y$ the logits, but $W$ the student head weight. The hyperparameter $\ell$ is the number of layers or depth. The MLPs are specified as

$$\mathrm{MLP}_l(v) = B_l \mathrm{ReLU}^2(A_l \mathrm{RMSNorm}(v) + b_l), \tag{76}$$

where $B_l \in \mathbb{R}^{m \times 4m}$, $A_l \in \mathbb{R}^{4m \times m}$, and $b_l \in \mathbb{R}^{4m}$. The teacher has the same architecture as the student, but has a larger depth $\ell^*$ in general. The loss is still the KL-divergence between teacher and student output distribution. We still use $m = 32$ and $n = 128$. Teacher has $\ell^* = 128$ and student has $\ell = 6$. The learning rate is 6e-4, and we train the student for 40000 steps. The code is in `exp-9.py`. After increasing the teacher inverse temperature $\beta^*$ we can observe $1/3$ time scaling in loss (Figure 24 and Figure 25). The aligned student analysis indicates that once the logits effectively follow gradient flow, we can have the $1/3$ time scaling without caring about model parameters leading to the logits. Our experiments on this generalized toy model support this intuition: the change in architecture does not affect the fact parameters follow gradient flow, or the low-temperature power-law behaviors of softmax and cross-entropy, and $1/3$ time scaling of loss should exist.

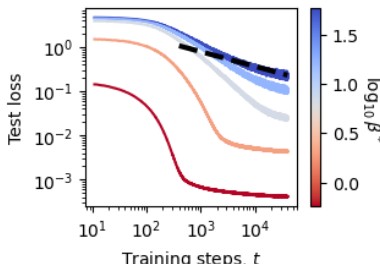

*Figure 24.* The generalized toy model exhibits power-law training behaviors at low temperatures. At high temperatures (small $\beta^*$), the loss vs. step $t$ curves are concave at first on the log-log plot, similar to exponential decay. Later, the loss saturates at a non-zero value due to the limitation of student depth ($\ell < \ell^*$). At low temperatures (large $\beta^*$), the loss vs. step $t$ curves converge to a line on the log-log plot later in training, like power laws with the same exponent. The dashed line refers to a power-law decay with exponent $1/3$.

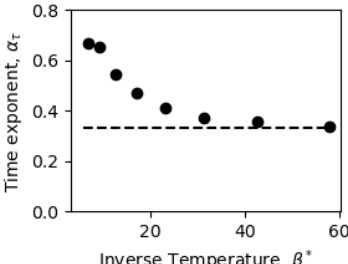

*Figure 25.* The generalized toy model exhibits power-law training behaviors at low temperatures. The losses of students at low temperatures are still far from the non-zero saturation value. We directly fit the final part as a power law with $t$. The exponent approaches $1/3$ when increasing $\beta^*$. The dashed line is $1/3$.

## D.8. MNIST Classification with a MLP

So far, we have tested various toy models, which all show $1/3$ time scaling. To test the generality of the scaling, we next try to study the training dynamics of networks on practical data or tasks.

We use the MNIST handwritten digit dataset, which consists of $D = 60,000$ training images of size $28 \times 28$ pixels, each associated with one of $n = 10$ class labels. Each image is flattened into a vector $x \in \mathbb{R}^{m_0}$ with $m_0 = 784$. To isolate training-time dynamics from stochastic effects of mini-batching, we load the entire training set and treat it as a single batch throughout all training steps, performing full-batch gradient descent. The experiment code is in `exp-4-2.py`, and the analysis is in `exp-4-2.ipynb`.

We parameterize the classifier as a multilayer perceptron (MLP) with $\ell$ hidden layers of uniform width $m$. Given an input $x \in \mathbb{R}^{m_0}$, the network computes a sequence of representations

$$h_0 = x, \tag{77}$$
$$h_l = \text{ReLU}\left(W_l\,\text{BN}(h_{l-1})\right), \quad l = 1, \ldots, \ell, \tag{78}$$
$$y = W\,\text{BN}(h_\ell), \tag{79}$$

where $W_1 \in \mathbb{R}^{m \times m_0}$, $W_l \in \mathbb{R}^{m \times m}$ for $l \geq 2$, $W \in \mathbb{R}^{n \times m}$, $y$ are output logits, and $\text{BN}(\cdot)$ denotes batch normalization applied before each linear transformation. The output weight matrix $W$ carries no bias term and is initialized to zero; all other parameters follow standard initialization. In this experiment, we set $\ell = 1$ and $m = 10$.

The model is trained to minimize the cross-entropy loss. Since each input has a probability of $1$ to be in one class, cross-entropy loss is equivalent to the KL divergence here. And the target distribution is certainly peaked: the temperature is exactly zero. We use the Adam optimizer with learning rate $\eta = 10^{-3}$ and default momentum parameters ($\beta_1 = 0.9$, $\beta_2 = 0.999$).

At each step $t$, we record three quantities: the training loss, the training accuracy, and the logit standard deviation averaged over input data. The mean logit standard deviation, or logit standard deviation for short, measures how confidently the

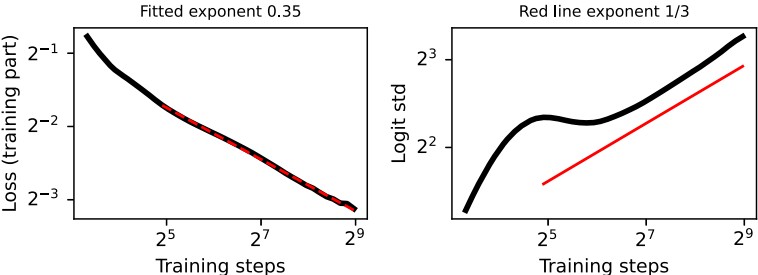

*Figure 26.* MNIST classification shows $1/3$ scaling in loss. (Left) The loss has a power-law region with fitted exponent 0.35. (Right) In a similar region where the loss is power-law, the logit standard deviation is also similar to a power law with exponent $1/3$ (compared to the red line whose slope or exponent is $1/3$).

network discriminates among classes on average, and is conceptually the same as inverse temperature. We use the symbol $\beta$ for inverse temperature in the main text for the toy model, specifically when the last hidden states are i.i.d. Gaussian.

We find that the loss has a power-law region and the fitted exponent is $0.35 \approx 1/3$ (Figure 26 left). Near the same region where loss is a power law, the logit standard deviation also enters a power-law growth region with an exponent similar to $1/3$ (Figure 26 right). These results, obtained from a different architecture solving a different task from toy models, suggest that the $1/3$ time scaling depends on generic properties like softmax and cross-entropy non-linearities and peaked output distributions, is independent of many other details, and should be general in real neural networks.

### D.9. LLM Data Analysis

Figure 6a is obtained by numerically calculating the entropy of the distribution obtained from $n = 32000$ i.i.d. Gaussian logits with standard deviation $\beta^*$. We scan $\beta^*$ from `torch.linspace(0, 9, steps = 20)`. For each $\beta^*$, we sample 1024 sets of logits (each set leads to a distribution and an entropy), and take the mean entropy. We use the vocabulary size $n = 32000$ from Hoffmann et al. (2022) since we use their fitting of irreducible loss (upper bound for entropy) 1.69. The value $c_0 = \sqrt{2 \ln n}$. With binary search, we know that the $\beta^*$ leading to entropy 1.69 should be between 6.6316 and 7.1053 (they are in `torch.linspace(0, 9, steps = 20)`). We take the average of 6.6316 and 7.1053 as a rough estimation of the intersection $\beta^*$ value, 6.87 in Figure 6. The code is in `test-6.ipynb`.

Methods of analyzing Pythia models are in Section C. The code is in `pythia-logit-i.ipynb` for $i = 0, 1, 2, 3$ (we analyzed four model sizes). We picked the checkpoints at steps 128, 256, 512, 1000, 2000, 3000, 6000, 12000, 16000, 32000, 48000, 79000, and 143000 for each model size. The steps are chosen such that they are roughly evenly spaced on the log scale. The raw loss as a function of step is plotted in Figure 27. The loss as a function of dynamic time $\tau$ is in Figure 6b. We obtained the dynamic time based on the learning rate scheduler of Pythia models (Biderman et al., 2023). Specifically, $\eta_t$ increases linearly from 0 to peak learning rate $\eta_{peak}$ for the first 1% steps. Then it decays as a cosine function,

$$\eta_{\text{peak}}[0.9(0.5(\cos(\theta_t) + 1)) + 0.1],$$

where $\theta_t = \frac{t - 0.01 t_{\max}}{0.99 t_{\max}} \pi$. We can obtain $\tau$ for its corresponding $t$ as

$$\tau(t) = \sum_{i=1}^{t} \eta_t. \tag{80}$$

For each model size, the last 11 points seem to be on one curve in Figure 6b, and the points before are on another curve. We fit the last 11 points with the format in Equation (20) for each model size in Figure 6b. The lines are drawn based on fitted parameters, which agree well with the data. Visually, more final points are on one curve in Figure 6b than in Figure 27. Our theory also suggests that the power laws should be seen with respect to $\tau$. We then fit the final points that are visually on one line with the format in Equation (20). The predicted loss based on fitted parameters agrees with real data well (Figure 6b). We further plot the Loss (training part), i.e., $L - L_{\backslash \tau}$, as a function of step $t$ in Figure 28, where the loss curves do not collapse. Once we plot $L - L_{\backslash \tau}$ with $\tau$ as in Figure 6c, the curves do collapse, supporting the correctness of Equation (20) –

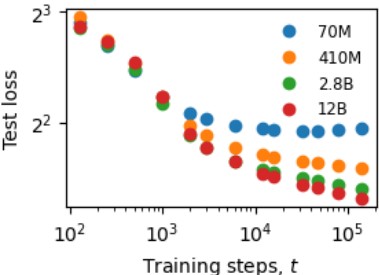

*Figure 27.* Raw loss as a function of training step. For each model size, the last 9 points seem to be on one curve, and the points before are on another curve.

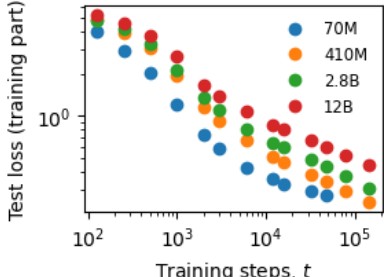

*Figure 28.* Loss (training part), i.e., $L - L_{\setminus\tau}$, as a function of step $t$. The value $L_{\setminus\tau}$ is obtained from the fitting described in Figure 6b. The loss curves are not collapsing.

there is a universal exponent $\alpha_\tau = 1/3$ and a constant $c_\tau$ independent of model sizes. Obtaining the logit standard deviation averaged over the evaluation dataset as described in Section C, we can directly plot it with the calculated $\tau$ in Figure 6d.

It may not be a fair comparison that $L - L_{\setminus\tau}$ from different models do not collapse with respect to $t$ but $\tau$ as $L_{\setminus\tau}$ is fitted with respect to $\tau$. We then also fit the raw loss with

$$L = \frac{c_t}{t^{\alpha_t}} + L_{\setminus t}, \tag{81}$$

separately for each Pythia model. We use the same set of data points used in the fitting with $\tau$. The results are in Figure 29 upper-right panel. We next plot $L - L_{\setminus t}$ in the lower-right panel of Figure 29. To quantify the quality of the collapse, we fit all $L - L_{\setminus t}$ data points from different models with a single line on the log-log scale, yielding $R^2 = 0.93$. As a comparison, we copied the fitting with $\tau$ (Figure 6b) to Figure 29 upper-left panel. All the $L - L_{\setminus\tau}$ values are fitted with $\tau$ with a line on log-log scale, whose $R^2 = 0.98$. Fitting loss with $\tau$ therefore has better collapse quality than fitting with $t$, supporting our theory that $\tau$ is the fundamental variable governing training dynamics.

In addition to Pythia models, we also evaluated Olmo models (OLMo et al., 2024), which is a more modern model family, to test the generality of the $1/3$ scaling. The code is in `olmo-logit-i.ipynb` for $i = 0, 1, 2$. We used the same dataset as above for evaluation. And we use the learning rate schedule reported in (OLMo et al., 2024) to calculate $\tau$ for the checkpoints evaluated. We fitted the loss with Equation (20), and plot $L - L_{\setminus\tau}$ with $\tau$ in Figure 30. For each model, we write the fitted $\alpha_\tau$ in brackets after the model size label (Figure 30), which are all close to be $1/3$. And loss curves $L - L_{\setminus\tau}$ with $L_{\setminus\tau}$ fitted separately for different model sizes collapse. The results are similar to those from Pythia models. We therefore can conclude that the $1/3$ scaling of loss is generic in LLMs.

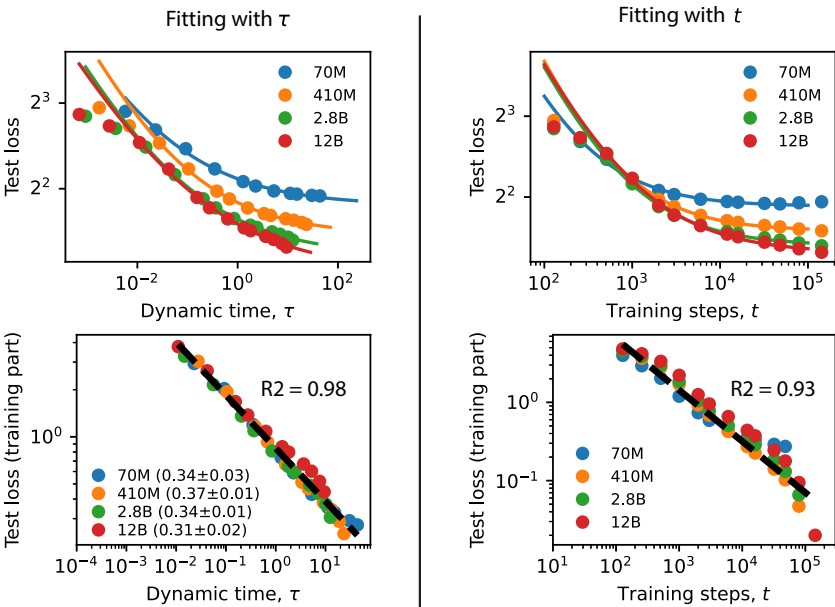

*Figure 29.* Comparison of loss fitting with $\tau$ and that with $t$ supports that $\tau$ is the fundamental variable controlling training dynamics. (Left panels) The upper panel is the same as Figure 6b. After fitting $L_{\setminus\tau}$ for each model, we fit $L - L_{\setminus\tau}$ altogether in the lower panel, yielding $R^2 = 0.98$, suggesting good line collapse. (Right panels) We fit the same set of data points with $t$ rather than $\tau$. After fitting $L_{\setminus t}$ for each model, we fit $L - L_{\setminus t}$ altogether in the lower panel, yielding $R^2 = 0.93$, which means worse collapse quality.

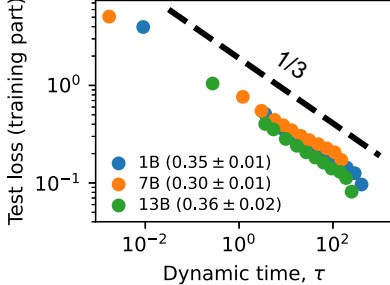

*Figure 30.* Loss (training part), i.e., $L - L_{\setminus\tau}$, as a function of dynamic time $\tau$ for Olmo-2 models (OLMo et al., 2024). The value $L_{\setminus\tau}$ is obtained by fitting the raw loss with Equation (20). Different colors refer to different models, labeled by model sizes, 1B, 7B, and 13B. The brackets contain the fitted exponents $\alpha_\tau$, which are all around $1/3$.

