# OpenReview forum: "Universal One-third Time Scaling in Learning Peaked Distributions"
_ICML.cc/2026/Conference — ICML 2026 regular_

### Official Review · Reviewer_P6db · 2026-02-27

**Soundness:** 3
**Presentation:** 3
**Significance:** 3
**Originality:** 4
**Overall Recommendation:** 5
**Confidence:** 3

**Summary:**

This paper investigates the origin of power-law loss scaling during LLM training. Using a minimal teacher-student toy model (with a single linear layer followed by softmax and cross-entropy loss), the authors show that peaked output distributions lead to power-law vanishing gradients, resulting in a loss decay of $L \sim \tau^{-1/3}$ in dynamic time $\tau$.
The derivation relies on a low-temperature expansion of the free energy and an empirical "aligned student ansatz", which is validated for the toy model. Together the dynamics is reduced to a scalar ODE. The authors argue that the 1/3 exponent derived from this model is universal, depending only on the softmax + cross-entropy structure rather than on data properties.
They validate the theory on the toy model and on Pythia models (70M–12B) evaluated on FineWeb, showing that fitted exponents follow near 1/3.

**Compliance With Llm Reviewing Policy:**

Affirmed.

**Final Justification:**

This paper suggests and demonstrates the novel idea that scaling exponents in LLM settings could be fixed at 1/3. With solid theory and empirical evidence, I believe this paper successfully contributes to the debate on the origins of scaling laws.

The authors have addressed all my concerns and questions, which made me increase my score to 5.

**Key Questions For Authors:**

Question 1: In practice, different scaling exponents can be observed depending on the model, task, and training setup. These can be attributed to (a) the model not yet being in the low-temperature regime, (b) the use of raw training steps rather than dynamic time, or (c) other factors? What does the procedure look like to distinguish these cases?

Question 2: The related works section is quite brief. The paper addresses prior work that attributes scaling laws to data structure, but does not sufficiently resolve the tension between those theories and the current one. How does the current theory relate to these prior explanations (e.g., Zipf distributions, manifold dimension)? Can these views be unified, e.g., by a proper rescaling of time? Are there regimes where multiple explanations hold?

**Limitations:**

Main limitations are adequately addressed in the Discussion section and across the main body.

**Strengths And Weaknesses:**

**Strengths**:

- The paper provides a clean mechanistic explanation for power-law loss scaling that does not require assumptions on the data structure. Instead, they identify the softmax + cross-entropy to govern how fast a model can reduce its errors. This is a significant conceptual contribution that shifts the focus from data properties to architectural components as the source of scaling laws.

- The toy model is well-designed: it isolates the softmax + cross-entropy mechanism in a minimal setup that is still analytically tractable. The aligned student ansatz is a very helpful empirical observation to solve the dynamics.

- The connection to statistical physics (universality, low-temperature expansions) is natural and provides useful intuition. The analogy to critical exponents gives a compelling conceptual framework to view the 1/3 scaling law as a universal property of the learning dynamics.

- The experimental validation on Pythia models across multiple scales (70M–12B) is a convincing demonstration that the mechanism is relevant beyond the toy setting. The fact that the fitted exponents consistently land near 1/3 is impressive.

- Besides the conceptual contributions, the paper is well written and clearly explains the main points of the theory and its implications. Figures are clear and informative and appendices are not overwhelming.

----


**Weaknesses**:


1. (Presentation, minor) The presentation of the theory is very intuitive for readers with a physics background. However, following the main body requires substantial physics intuition (e.g., free energy, partition functions, low-temperature expansions). It would help to provide clearer guidance on which appendix sections to consult for readers less familiar with statistical mechanics.

2. (Experimental gap) One important aspect of the theory is the independence of the 1/3 exponent from the data structure. While this follows from the derivation, it is a crucial claim that should also be verified experimentally in the toy model. E.g., by varying the input distribution or the teacher weight structure and confirming that the exponent remains unchanged.

3. (Scope) The 1/3 scaling applies only in the intermediate regime where $\beta$ is large but $\beta \ll \beta^*$. The paper clearly identifies the two regimes (exponential vs. power-law) in the toy model and provides a threshold $\beta = c_0$ separating them. However, for real LLMs, the temporal crossover within a single training run is not characterized: at what point during Pythia training does the 1/3 regime kick in, and how much of the training budget does it actually govern? Some discussion of this crossover and its practical implications would be helpful.

4. (LLM validation, major) The validation on Pythia models is the main strength of the paper. One essential aspect allowing for this connection is the transformation from training steps to dynamic time $\tau$. Even in the Appendix (line 1242 - 1252), this transformation is not fully described. Assuming the theory is correct, the exponent is universally 1/3 in dynamic time, but the transformation tells us how this relates to the exponent we observe in raw training steps. This transformation therefore requires more explanation. In particular:
    - **4.1** The paper should explicitly describe how one obtains the loss as a function of dynamic time for the Pythia models. If people want to check the theory on their own models, a clear and reproducible procedure for the transformation should be provided.
    - **4.2.** Including a fit in raw training steps alongside the dynamic time fit (in Figure 27) would show how much the transformation affects the observed scaling.
    - **4.3.** Reporting fit quality metrics (e.g., $R^2$) for both $t$ and $\tau$ would quantitatively support the claim that dynamic time is the right variable.

5. (LLM validation) The argument that real next-token distributions are sufficiently peaked relies on an indirect mapping: measured entropy is converted to an equivalent inverse temperature assuming i.i.d. Gaussian logits, which is then compared to the threshold $c_0$. However, real LLM logits are not i.i.d. Gaussian, and peakedness varies significantly across contexts (e.g., highly peaked after deterministic continuations, quite flat after open-ended prompts). The theory assumes uniform peakedness, and it is not addressed how averaging over a mixture of peaked and flat distributions affects the exponent. A natural experiment would be to train a student against a mixture of teachers with different $\beta^*$ values (some peaked, some flat) in the toy model and check whether the 1/3 exponent still holds.

I believe this is a very good piece of work, relevant to the community. I am happy to raise my score if the authors address the points above, in particular Weakness 4.

---

> ### Author Rebuttal · Authors · 2026-03-30
>
> We thank the reviewer for the comprehensive summary, detailed comments, and thoughtful suggestions. We will address the concerns in the following.
>
> **Weakness 1**
> Great suggestion. We are also considering removing some terminologies to reduce the barrier for ML readers.
>
> **Weakness 2**
> Important point! We have added experiments making eigenvalues of $ W^{T\star}W^{\star} $ power-law and varying the exponent. The inputs $x$ are still iid Gaussian, but logits $y$ now have power-law covariance. We find that changing the power law exponent in data does not change the exponent $1/3$ but coefficient (Fig3 in the [link](https://anonymous.4open.science/r/time-E3D7)). We will further try adding structures to the inputs $x$. There are conceptual difficulties we need to solve: what does the commutability of $x$ covariance and $y$ covariance mean? We believe that by adding the experiments, the completeness of our paper is improved.
>
> **Weakness 3**
> Thank you for asking. In Fig. 6b, one can see that the loss starts to be $1/3$ scaling after $\tau \approx 10^{-2}$ which is usually step 1000. We think the majority of training is spent in the 1/3 scaling regime, since the total number of steps is usually above 100k. However, we are not able to derive a bound for LLMs. Before the 1/3 scaling regime, the loss decay may be more related to the data power law. At first, LLMs are data-limited, but very soon they become step-limited. We will discuss these points in the paper.
>
> **Weakness 4**
> * In the Pythia paper, the learning rate $\eta_t$ is a function of step $t$. Specifically, $\eta_t$ increases linearly from 0 to peak learning rate $\eta_{peak}$ for the first 1% steps. Then it decays as a cosine function, $\eta_{peak}[0.9(0.5(\cos(\theta_t)+1)) + 0.1]$, where $\theta_t = \frac{t - 0.01t_{\max}}{0.99 t_{\max}}\pi$. As explained in our paper, we can obtain $\tau$ for corresponding $t$ as $\tau(t) = \sum_{i=1}^t \eta_i$. We will add to our paper.
> * Thanks! We added the fitting using the same points but with respect to $t$ (See Fig4 in the [link](https://anonymous.4open.science/r/time-E3D7)). The fitted exponents $\alpha_t$ are larger, around $0.6\sim 0.8$.
> * We fit the training part of the loss (a linear fit in log-log). The $R^2$ for $\tau$ is $0.98$ while the $R^2$ for $t$ is $0.93$. This suggests that $\tau$ is the fundamental variable governing training dynamics.
> * We also added evaluation on OLMo-2 models, where the fitted $\alpha_\tau$ are around $0.30\sim 0.35$ (Fig1 in the [link](https://anonymous.4open.science/r/time-E3D7)). This further supports our conclusion.
>
> **Weakness 5**
> This weakness is critical, yet not easy to address for LLMs. We appreciate the suggested experiments, which are super interesting. In LLMs, depending on the context, the output entropy can be very different. In our toy model, the analogy is that for different inputs $x$, the output entropies can be different. A mixture of teachers can lead to this property. But we think there is a cleaner way: introducing structures of $W^*$ as described in the reply to Weakness 2 above. The structure breaks the uniformity of output entropies: some directions will have larger norms and smaller output entropy. We found that the $1/3$ scaling still holds. Future work is needed to systematically study this case.
>
> **Q 1**
> In our view, one should start analysis by using $\tau$, which naturally avoids (b). For LLMs, usually after $\sim 10^3$ steps, the training is in low-temperature regime. It may also be important to check the entropy of outputs to know whether the model is in the low-temperature regime. If one cannot observe the $1/3$, it may be due to a large learning rate/small batch size, which makes noise dominate.
>
> **Q 2**
> * Great question. We will try to re-organize the text or to add to appendix. Our mechanism cannot be unified with previous ones. The heart of previous theories is that "more important features are learned first". Mathematically, it is true for MSE where each feature converges exponentially with a characteristic time scale. Importance is reflected in the time scale. If the time scale distribution follows a power law, we can have power-law scaling of loss with training. Yet, our mechanism suggests that Softmax/CE leads to power laws by themselves. Each feature will converge following a 1/3 power law, and the importance is reflected in the coefficient. The overall loss summing over losses of different features is inevitably a 1/3 power law, regardless of importance/coefficients distribution.
> * The data manifold argument uses a different setup. They imagine that models are trained to the optimal on training data (training loss can be zero), and study the scaling of the test loss. We use online learning here, where each training data point is seen once, and the model is not trained to optimality on training data (training loss is super close to test loss).
> * Previous theories are either in different regimes or have different setups.

---

> > ### Author Rebuttal · Reviewer_P6db · 2026-04-01
> >
> > Thanks for the detailed response. The authors have addressed all my concerns and questions. I believe this is a very good and relevant piece of work. I am raising my score to 5 and hope to see it published.
> >
> > Out of interest: According to your theory, to improve compute scaling we should either look for better loss functions or "warp" time more efficiently. Regarding the latter, can the theory provide guidance on how to warp time optimally, e.g., by constraining the learning rate schedule?

---

> > > ### Author Response · Authors · 2026-04-04
> > >
> > > We thank the reviewer for the support. The suggestions are very helpful, which we believe will improve the completeness and presentation of our work a lot.
> > >
> > > We are also very interested in the learning rate scheduler and, in general, the choice of learning rate. Based on the current theory, considering a perfect noiseless gradient, the learning rate scheduler does not matter once we change into $\tau$ as long as the learning rate is low, and then the continuous limit is correct.
> > >
> > > We believe that a learning rate scheduler is needed to handle gradient noise. In our mind, there are two places where gradient noise is important:
> > > * The first mechanism is related to the river valley picture of the loss landscape (arxiv:2410.05192). There are sharp directions and flat directions. The parameters are easy to reach equilibrium along sharp directions, and the loss along these directions is mainly due to noise. Decreasing the learning rate will reduce the noise and decrease the loss along these directions. A good scheduler should balance the annealing on the sharp directions and the decay along flat directions. Our understanding is that the sharp directions are in Transformer layers, and we need to extend the current toy model to study this perspective of the learning rate scheduler.
> > > * The second perspective (which was mentioned in Discussion) is that a large learning rate prevents momentum accumulation when noise exists and the gradient is vanishing, and therefore hinders rotation. A large learning rate is desired to increase the norm fast, yet a small learning rate is preferred to facilitate rotation in the presence of noise. We need to study the signal-to-noise ratio when the weight norm increases, which may decide the optimal learning rate. Our current toy model is ideal for studying this second perspective.
> > >
> > > Related to the choice of optimal learning rate, we saw one recent manuscript (arxiv:2603.28743) arguing that for both Muon and Adam, the optimal (peak) learning rate should scale as $D^{-0.32}$. This 0.32 is so close to 1/3 that we feel it is not a coincidence. There might be some deep connection to our 1/3 scaling mechanism, and we may find the connection after considering noise and the extra loss due to noise. We believe that our paper will open a venue for a range of future studies.

---

### Official Review · Reviewer_ouP1 · 2026-03-11

**Soundness:** 2
**Presentation:** 3
**Significance:** 1
**Originality:** 3
**Overall Recommendation:** 3
**Confidence:** 4

**Summary:**

The paper proposes an extremely simple scenario (Ansatz) where one can demonstrate that the loss of an LM as a function of training time follows an exponential with exponent 1/3. Similar exponents are then observed on a toy model and on the Pythia family. This exponent is then claimed to be "Universal".

**Compliance With Llm Reviewing Policy:**

Affirmed.

**Final Justification:**

My criticism might apply to all scaling theories but I still feel that the Ansatz is too simple. Basically if you have the solution of a simple problem (single degree of freedom), you show 1/3 convergence, which is shared with the model you tried in practice. I do not see why it would apply to a more complex model.

In most neural LM, the last layer norm has a scale vector with very few parameters (dim_model) that can be seen as effectively controlling the scale of the last matrix (temperature). I feel that even if it would be free to adjust this vector to the current optimum (given everything else constant), learning the other parameters of the network (which you do not study) would still exhibit a power-law convergence. This means that the other component also follow a similar power-law behavior without theoretical justification.

For Q3, are the last matrix vectors close to Gaussian in practice? I would expect that rare/common token types would have very different norms.

The empirical support is nice, I appreciate that you have included some versions of Olmo. I increased my score.

**Key Questions For Authors:**

Q1: In the toy example, do you also get ⅓ if you add noise (i.e. intrinsic uncertainty) and have the teacher and student belong to different function families? E.g. the teacher is a mixture of linear+softmax?

Q2: Could improve notation, p(x), q(x) are used to denote p(y|x), q(y|x), a more common notation.

Q3: Do you need W^\star and x to be normal, is it a strong constraint, realistic?

Q4: Why do you extract data dependent constants upper bound for next-token distribution entropy from Chinchilla if your study is on the Pythia dataset?

Q5: “The critical roles of architectural factors, i.e., softmax and cross-entropy, in shaping training dynamics.” -> do you have evidence that this would not happen with other network transfer functions or loss functions?

Q6: The training runs for Olmo, Olmo2 and Olmo3 are also public, are all their exponents also around ⅓?

**Strengths And Weaknesses:**

Strengths
- The paper reads well, has sufficient references
- The paper is technically sound.

Weaknesses
- I have a hard time assessing the impact the paper could have. The only proof with ⅓ exponent is for the Ansatz and there is no guarantee that the exponent would be close or bounded by this value in the real case. The Ansatz is way too simple, everything about a very simple function is known except a single scalar and there is not even noise in the training data. I have a hard time seeing how such a paper could influence future research or practical decisions. The main motivation in the intro is that the other papers “ Despite the accuracy and predictive power of these fitted power laws, they remain largely empirical.” I would say that the Ansatz is the only non-empirical part here as well and it does not share much with the problem of learning an LLM. The term “Universal” is a misnomer, nothing prove that there is a universal ⅓ constant in practice.

---

> ### Author Rebuttal · Authors · 2026-03-30
>
> **"I have a hard time assessing the impact the paper could have"**
> Neural scaling laws are one of the most important empirical results that have shaped the success of LLMs. We proposed a novel mechanism for the origin of the scaling laws, which has theoretical insights and empirical support. If our mechanism is correct, it can lead to improvement of neural scaling as suggested in Sec 6. "Understanding why training loss follows power laws is central to scaling-law science" (iL9t), and our proposed mechanism is "fresh angle" (iL9t), "new as main strength" (SGZ3), "clean" (P6db) and "a significant conceptual contribution" (P6db), which "may serve as a basis for several expansions and the extensions of the idea, significantly enriching ourunderstanding of power-law origins in LLMs" (SGZ3).
>
> **"The only proof with ⅓ exponent is for the Ansatz and there is no guarantee"**
> This is true, yet this criticism applies to all scaling theories. And $1/3$ is the exact solution under the ideal case, which is well supported by a range of real experiments. Our original LM head toy model (Sec. 3), toy model with MLP (Sec. D6), toy model with weight decay (Sec 3), toy model with data power law (new experiments, Fig3 in [link](https://anonymous.4open.science/r/time-E3D7)), MNIST classification via MLP (new experiments, Fig2 in [link](https://anonymous.4open.science/r/time-E3D7)), Pythia (Sec. 4), Chinchilla, OLMo (new experiments, Fig1 in [link](https://anonymous.4open.science/r/time-E3D7)) all show 1/3 scaling.
>
> **"The Ansatz is way too simple"**
> We view this as a strength, not a weakness. We want to find the fundamental origin of the power law. The simplicity allows us to extract insights (Softmax/CE produce power laws themselves). More complicated situations cannot be solved, but we did use experiments to show that 1/3 still holds. Noise is irrelevant for the emergence of power law. One can study how noise affects the scaling, which is incremental and not the focus of this paper.
>
> **"I have a hard time seeing how such a paper could influence future research or practical decisions."**
> Our theory has concrete implications for LLM training and architecture. First, training LLMs should avoid large weight decays as LLMs need to reach large logit values for peaked distributions. For training, we hope to have new optimizers strong at rotating parameters. If the optimizer can outperform the gradient flow, we may have faster loss scaling. Finally, future architectures may make use of the token hierarchy to avoid outputting a distribution over the whole vocabulary.
>
> **"The term “Universal” is a misnomer, nothing prove that there is a universal ⅓ constant in practice."**
> We use the term "universal" following "universality" in physics. It means that the exponent 1/3 is independent of other details once the temperature is low. We emphasized the premise/condition in the main text. Our original LM head toy model (Sec. 3), toy model with MLP (Sec. D6), toy model with weight decay (Sec 3), toy model with data power law (new experiments), MNIST classification via MLP (new experiments), Pythia (Sec. 4), Chinchilla, OLMo (new experiments) all show 1/3 scaling. These real experiments prove that 1/3 scaling is practical.
>
> **Q1** In our toy model, the data are intrinsically uncertain because the teacher outputs a distribution. If student cannot learn the teacher perfectly, as long as Softmax/CE is used, and the distribution is peaked, the loss will converge to a non-zero value following the 1/3 power law. If the purpose is to introduce gradient noise in training, we have addressed this partly in the main text by increasing learning rates. We also tested different batch sizes. For small batch sizes (e.g. 128), the loss scaling can be slightly slower (e.g., 0.31).
>
> **Q2** In our notation, $p(x)$ is a vector, and $p_i(x) := p(i|x)$ is the correct notation, not $p(y|x)$.
>
> **Q3** It is required for the analytic analysis. Otherwise, we cannot solve (and probably no one can). But we added experiments to study power-law $W^*$, which still shows 1/3 scaling. See our reply to Weakness 2 of P6db.
>
> **Q4** Chinchilla used around 400 points to do the fitting, whose irreducible loss value is then more accurate. We can do our own fitting based on Pythia models and different datasets (c4, wikitext, bookcorpus, and pile). The irreducible loss values are in the range 1.5 to 2.2 with a large uncertainty, which are close to the Chinchilla value, and are all in the low-temperature regime.
>
> **Q5** MSE does not have this property, which is well-known and is the reason previous works need to assume power laws in data to obtain power laws during training. You can read the famous arXiv:1312.6120. See our Sec. A4, where we show CE can be approximated by MSE in the high-temperature regime, and there is no power law.
>
> **Q6** Given the short time, we have only evaluated OLMo 2 models, which do show 1/3 scaling. See Fig1 in [link](https://anonymous.4open.science/r/time-E3D7).

---

> > ### Author Rebuttal · Reviewer_ouP1 · 2026-04-07
> >
> > My criticism might apply to all scaling theories but I still feel that the Ansatz is too simple. Basically if you have the solution of a simple problem (single degree of freedom), you show 1/3 convergence, which is shared with the model you tried in practice. I do not see why it would apply to a more complex model.
> >
> > In most neural LM, the last layer norm has a scale vector with very few parameters (dim_model) that can be seen as effectively controlling the scale of the last matrix (temperature). I feel that even if it would be free to adjust this vector to the current optimum (given everything else constant), learning the other parameters of the network (which you do not study) would still exhibit a power-law convergence. This means that the other component also follow a similar power-law behavior without theoretical justification.
> >
> > For Q3, are the last matrix vectors close to Gaussian in practice? I would expect that rare/common token types would have very different norms.
> >
> > The empirical support is nice, I appreciate that you have included some versions of Olmo. I increased my score.

---

> > > ### Author Response · Authors · 2026-04-07
> > >
> > > ## "I do not see why it would apply to a more complex model."
> > >
> > > Because the argument of Taylor expansion of the CE loss and gradient descent dynamics will be the same for more complex situations. And if we keep the leading terms only, the scaling should not change. Yet, the rigorous analytical proof is hard to obtain. That is exactly why we did so many experiments, ranging from our simple toy models to toy models with data structures, to toy models with weight decay, to toy models with MLP, to MLP for MNIST classification, to LLMs (OLMo, Pythia). And our prediction also agrees with Chinchilla scaling laws. In a lot of empirical cases we tried, our ansatz is wrong, yet the $1/3$ scaling persists, supporting our argument.
> > >
> > > If a mechanism can only be established by solving the real model or all complex situations, then no mechanism can be established. The purpose of our toy model is to understand the insights. For your information, Einstein only solved a **one-dimensional** case in his original paper on Brownian motion.
> > >
> > > Mathematicians and physicists may spend the next few decades trying to solve our toy model in more complex situations or different variations of our toy models, yet before all those solving, some people like us need to propose/show it first.
> > >
> > > ## "learning the other parameters of the network (which you do not study) would still exhibit a power-law convergence"
> > >
> > > See our Sec. D6, where we **did** study the training of other parameters (not just the head but also some MLP before the head). And we did discuss some interpretation in the main text.
> > > - One perspective is that as long as the logits effectively follow gradient flow dynamics, the $1/3$ scaling should hold, regardless of training head or other parameters.
> > > - Another perspective is that other parameters also will have power-law vanishing gradient due to the Softmax and CE loss bottleneck.
> > >
> > > Again, we agree that one of the weaknesses of our paper is that we cannot solve all these complex situations, and it is interesting to study these cases in future work. But it is nearly impossible to require the very first paper to solve all the things analytically.
> > >
> > > ## For Q3, are the last matrix vectors close to Gaussian in practice?
> > >
> > > We added a power-law spectrum to $W^*$. So, yes, rows along different directions will have very different magnitudes. But the $1/3$ scaling is still there (See details in our reply to Weakness 2 of P6db and [Fig3 in the link](https://anonymous.4open.science/r/time-E3D7))! The reason is that the data structure will change the coefficients of the Taylor expansion, but not the exponent. The core idea of universality is that these details do not matter for the exponent.

---

### Official Review · Reviewer_iL9t · 2026-03-13

**Soundness:** 3
**Presentation:** 3
**Significance:** 3
**Originality:** 3
**Overall Recommendation:** 5
**Confidence:** 3

**Summary:**

This paper shows that the power-law decay of training loss in LLMs can arise from the softmax and cross-entropy rather than from power-law structure in the data. Using a teacher–student toy model, the authors derive a $L \sim \tau^{-1/3}$ scaling in a low-temperature regime under an aligned-student assumption, and present toy experiments and Pythia checkpoint analyses that they claim support this mechanism. The central claim is that learning highly peaked next-token distributions creates a generic optimization bottleneck that yields a universal $1/3$ scaling exponent.

**Compliance With Llm Reviewing Policy:**

Affirmed.

**Final Justification:**

I believe my concerns and questions are resolved, so I increase my score to 5.

**Key Questions For Authors:**

1. Could the authors provide a more explicit justification for the low-temperature expansion in Eq. (13), in particular why the free energy should be analytic in $(1/\beta)$ in this setting?

2. Have the authors checked whether the same $1/3$ behavior appears in another public LLM family beyond Pythia, or under a controlled change of output entropy in the toy model?

**Limitations:**

yes

**Strengths And Weaknesses:**

**Strengths**
* Understanding why training loss follows power laws is central to scaling-law science, and the focus on output-layer nonlinearities is a fresh angle.
* The paper is generally readable and easy to follow. The organization from toy model, to theory, to toy experiments, to LLM evidence is logical.
* The main derivation is conceptually elegant. Equations (15)–(19) provide a clear and memorable chain of reasoning: a low-temperature free-energy correction yields $L \sim \beta^{-1}$, the gradient dynamics imply $\frac{d\beta}{d\tau} \sim \beta^{-2}$, which leads to $\beta \sim \tau^{1/3}$ and therefore $L \sim \tau^{-1/3}$.

**Weaknesses**
* The empirical support for the connection to real LLMs is currently too limited. Section 4 considers only Pythia checkpoints and a single evaluation dataset. Although Figure 6b is intriguing, evidence from a single model family is not sufficient to justify the broader claim that this mechanism explains LLM scaling behavior. This is a central concern, since the paper’s significance depends on whether the toy-model mechanism genuinely transfers to realistic LLM training.
* While I am not an expert in this area, I found the toy-model analysis interesting and potentially insightful, but I was not fully convinced by one of the central steps in the argument. In particular, the paper argues that in the low-temperature regime, a free-energy expansion yields $L \sim \beta^{-1}$ and $-dL/d\beta \sim \beta^{-2}$, which in turn implies $\beta \sim \tau^{1/3}$ and $L \sim \tau^{-1/3}$ (Eqs. 15--19). My concern is that this conclusion appears to depend quite heavily on asymptotic assumptions that are not sufficiently justified in the current presentation, especially the validity of the low-temperature expansion and the existence of an intermediate regime where $\beta$ is large but still satisfies $\beta \ll \beta^*$.

---

> ### Author Rebuttal · Authors · 2026-03-30
>
> We thank the reviewer for the important questions, which are very to-the-point. We will address them in the following.
>
> **Weakness 1**
> There are many public LLMs, but not many have published checkpoints. We added evaluations of OLMo-2 1B, 7B, and 13B models, and the measured exponents are within $0.30\sim 0.35$ (Please see Fig1 in [link](https://anonymous.4open.science/r/time-E3D7)), supporting our conclusion. The fact that Chinchilla scaling laws have an exponent $0.28$ also supports us. We also added experiments using MLP to do MNIST classification, where we can also observe the $1/3$ scaling (Fig2 in [link](https://anonymous.4open.science/r/time-E3D7)). We believe that our mechanism genuinely exists in LLMs and even everywhere in ML.
>
> **Weakness 2 and Question 1** Thank you very much for pointing this out. We provided a mathematical explanation in reply to Weakness 2 of SGZ3. The key point is that the CDF of logit differences (energy gaps) should be analytic, which is a mild assumption. We will add the discussion to the paper. In addition to math, we think our main text Fig. 3 provides numerical evidence for the validity of low-temperature expansion.
>
> **Question 2** We partially addressed this in the reply to Weakness 1 above. We are not sure about the meaning of "controlled change of output entropy in the toy model". Do you mean we control the magnitude of logits and make the student output entropy unchanged? We think Fig. 5, c and d already partly address this question, where we have weight decay and $\beta$ does not change much at the end. However, due to rotation, the loss can still decrease, which also follows the $1/3$ scaling.

---

> > ### Author Rebuttal · Reviewer_iL9t · 2026-04-01
> >
> > Thank you for your response. It has addressed my concerns, and I have no further questions.

---

> > > ### Author Response · Authors · 2026-04-04
> > >
> > > Thank you very much for your suggestions, questions, consideration, and support!

---

### Official Review · Reviewer_SGZ3 · 2026-03-24

**Soundness:** 3
**Presentation:** 3
**Significance:** 4
**Originality:** 4
**Overall Recommendation:** 5
**Confidence:** 3

**Summary:**

The paper proposes a new mechanism for the power-law decay of LLM training loss with time. Unlike prior work that attributes power laws to power-law structure in data, the authors argue that the combination of softmax nonlinearity and cross-entropy loss is sufficient: when the target distribution is peaked (low temperature), the loss vanishes as power laws in the inverse temperature $\beta$, which grows as $\tau^{1/3}$ under gradient flow, giving $L \sim \tau^{-1/3}$. The analysis uses a one-layer teacher-student toy model mimicking the LM head. Under an "aligned student ansatz" ($W \propto W^*$), the weight dynamics reduce to a scalar ODE for $\beta$, solved via low-temperature expansion of the free energy from statistical mechanics. The authors validate the theory on the toy model (SGD, Adam, various initializations, weight decay) and show that Pythia models (70M--12B) exhibit exponents close to $1/3$ when loss is plotted against dynamic time $\tau$.

**Compliance With Llm Reviewing Policy:**

Affirmed.

**Key Questions For Authors:**

**General target distributions.** In LLMs the ground-truth next-token distribution $p_i(x)$ is not generated by a softmax teacher with the same architecture. What happens when the target is a general distribution, not realizable by the student? One key difference is non-zero KL divergence at the optimum. Does the $1/3$ exponent survive, and if so, does it describe convergence to the irreducible loss?

**Limitations:**

No negative societal impact. The authors acknowledge some of the limitations of their model, though some other remains less acknowledge (see weaknesses).

**Strengths And Weaknesses:**

### Strengths

1. The main strength of the paper is the proposed mechanism. To the best of my knowledge, this is a new mechanism; it goes beyond the domminant data-based narrative in power-laws origin; it is relatively simple as it is based on just on Taylor expansion making it universal as it does not require any specific details or assumptions about the problem. This is a clean and surprising insight, and alone it outweighs many smaller concerns in this review. I also believe that this model may serve as a basis for several expansions and the extensions of the idea, significantly enriching ourunderstanding of power-law origins in LLMs.

2. The authors also discuss the ways their toy model might be wrong, and what happens in this case. Main example of that is rotational vs radial degrees of freedom in the weight decay experiments. Discussion section also points to a few good directions.

3. The experimental coverage is thorough. The toy model is tested across three orders of magnitude in $\beta^*$, multiple optimizers (SGD, Adam), learning rate schedules, initialization scales, and weight decay. The Pythia experiments provide a reasonable bridge to real LLMs.

### Weaknesses

1. **The proposed mechanism likely coexists with other mechanisms, and the paper does not discuss this interaction.** Representation learning is central to LLM training — the model must learn *which* tokens are correct, not just become more confident about a fixed ordering. Consider MMLU as a concrete example: the benchmark tests which of the A/B/C/D tokens has the largest logit. Only the ordering matters; the magnitude is irrelevant. Yet MMLU scores improve steadily throughout pretraining, meaning the model is learning to reorder logits — something the proposed $1/3$ mechanism cannot capture, since under the aligned ansatz the ordering is correct from initialization.

This suggests the identified softmax/CE bottleneck coexists with other mechanisms (e.g., representation learning in transformer layers) that drive capability acquisition and may have their own power-law character. Assuming some kind of superpostion of different mechanisms in the total loss trajectory, the observed exponent would be determined by the slowest-decaying component. If the $1/3$ mechanism is the bottleneck, it puts an upper bound on the convergence exponent. But if another mechanism is slower (exponent $< 1/3$), that one dominates instead — and this may be what happens in practice: the Chinchilla fitted exponent of $0.28$ is below $1/3$, which the paper explains via a nonlinear $\tau$-vs-dataset-size mapping, but could alternatively reflect a slower mechanism being the actual bottleneck.

The paper claims in Section 4 that architectural components "may not matter" and shows the mechanism persists with additional MLP layers (Section D.6), but this does not address the key issue: the toy model has no representation learning (inputs $x$ are i.i.d.), while in LLMs the hidden states change throughout training. A more careful discussion of how the $1/3$ mechanism interacts with representation learning would significantly strengthen the paper.

2. **The analyticity of free energy in $\beta^{-1}$ needs justification.** The entire derivation of the $1/3$ exponent rests on the Taylor expansion $F(\beta) = -c_0 - c_1\beta^{-1} - c_2\beta^{-2} + \cdots$ (eq. 13/46). But the partition function $Z(\beta) = \sum_i e^{-\beta\epsilon_i}$ is naturally a sum of exponentials in $\beta$, so it is analytic in $\beta$ — it is not obvious from where the analyticity in $1/\beta$ comes from. I suspect this is a known result in statistical physics, but for the ML audience it is a non-trivial step that the paper does not explain. Since this expansion is the engine behind the main result, the paper should provide either an explicit derivation or a reference to a rigorous justification.

3. **The "universality" is narrower than the presentation suggests.** The $1/3$ exponent holds only in the intermediate regime: $\beta$ must be large enough to be in the low-temperature expansion, yet much smaller than $\beta^{\star}$. The paper acknowledges this but the framing — the title says "universal," the abstract says "universal exponent" — overstates the scope. In particular: (a) high-temperature data gives exponential, not power-law, convergence; (b) near convergence ($\beta \approx \beta^\star$), the exponent changes; (c) the mechanism requires that $\beta^* > c_0 \approx \sqrt{2\ln n}$, which the paper verifies for Pythia only indirectly via an i.i.d. Gaussian proxy for the entropy bound. It would be more honest to frame this as "the intermediate regime generically produces $1/3$" rather than "universal $1/3$."

4. **Weight decay contradictions.** in Section 6, the authors suggest training should "avoid large weight decays" since the mechanism relies on norm growth. But weight decay is empirically essential for stable LLM training, and the paper's own experiments (Figure 5c-d) show $1/3$ scaling persisting under weight decay via rotation — a regime the theory does not cover analytically.

---

> ### Author Rebuttal · Authors · 2026-03-30
>
> We thank the reviewer for the accurate summary, thoughtful comments, and support. We will address the concerns in the following.
>
> **Weakness 1**
> - Both rotation (which token is correct) and increasing norm (be more certain) are important for decreasing loss. Our theory only proves that the norm increase leads to $1/3$ scaling (Result 1), yet our experiments show that rotation can also lead to $1/3$ scaling (Result 2) in the low-temperature regime. The center of the $1/3$ scaling is power-law vanishing loss/gradients, which seems to be true for all directions. However, more efforts are needed to prove the rotation case, which we are trying.
> - Thank you for this suggestion! The scope of our paper is to isolate, propose, and understand the Softmax/CE mechanism. The interaction and coexistence of mechanisms are more realistic and worth discussing, for which we do not really have a good answer now, but we are happy to list all possibilities. Arxiv:2404.10102 tried to reproduce Chinchilla scaling laws and measured an exponent $0.37$, which is above $1/3$. The uncertainty of fitting is also one source of the difference we should discuss.
> - In Sec D6, the toy model with MLPs does have hidden states changing throughout training. Our guess is that in LLMs, as long as the logits are following gradient descent effectively, we will have the same $1/3$ power-law scaling. Lazy learning may prevent logits from following gradient effectively, which may lead to a slower loss decay. The low-temperature regime we are exploring is something completely new conceptually. It is important for the whole field to understand its connection/difference with/from previously explored regimes.
> - If the reviewer thinks "learning more important features first" is a consequence of "representation learning", we argue that this may be correct only in the high-temperature regime, where the CE loss can be approximated by MSE loss (Sec. A4). Each feature is learned in an exponential manner, and more important ones have higher convergence rates. In the low-temperature regime, we think that the loss associated with each feature is a 1/3 power-law whose coefficient is determined by the feature's importance. We cannot define a characteristic time scale of being learned for learning each feature. We are working on the math.
>
> **Weakness 2**
> We thank the reviewer for this question. It is natural in physics but indeed very subtle in math. Let's use binary classification ($n = 2$) to illustrate the idea. $\beta F = -\langle \ln(e^{-\beta\epsilon_{1}} + e^{-\beta\epsilon_{2}}) \rangle = \beta \langle \min(\epsilon_{1}, \epsilon_{2})\rangle - \langle e^{-\beta \Delta \epsilon}\rangle - \cdots$ where we used Taloy expansion of $\ln$ when $\beta$ is large, and $\Delta \epsilon>0$ is the energy gap. $\langle e^{-\beta \Delta \epsilon}\rangle = \int_0^{\infty} e^{-\beta \Delta \epsilon} d\rho(\Delta \epsilon) \approx \int_0^{1/\beta} d\rho(\Delta \epsilon) = f_{\Delta \epsilon}(\beta^{-1})$, where $f_{\Delta \epsilon}$ is the CDF of $\Delta \epsilon$. The rigorous argument is that if energy gaps have analytic CDFs, then $F$ can be well approximated by an analytic function of $\beta^{-1}$. And for that analytic function, we can do a Taylor expansion around $\beta^{-1} = 0$ and get the power-law scaling with $\beta$.
>
> **Weakness 3**
> We thank the reviewer for this suggestion. We will tune our conclusions. In the title, we did use "in learning peaked temperature" to address your point (a). Throughout the paper, we will emphasize the "intermediate regime". We use the word “universal” to echo "universality" in statistical physics, which does not mean that the 1/3 time scaling is true for all possible training cases.
>
> **Weakness 4**
> We agree that for current LLMs, weight decay is important for stable training. That's why we suggested other methods in Discussion to strengthen stability while removing weight decay. For our toy model, we use weight decay as a probe to scientifically study training dynamics with restricted norms. We are working on the math when the aligned student ansatz is wrong.
>
> **General target distributions**
> Thank you for asking! Combining model size scaling and training time scaling is what we are working on. We have tried training a student model with a smaller width than the teacher. Preliminary results show that the loss converges to a non-zero value following the $1/3$ scaling, which is also the case for actual LLMs. However, our theoretical analysis of this case is still ongoing.

---

> > ### Author Rebuttal · Reviewer_SGZ3 · 2026-04-03
> >
> > I thank the authors for their response and maintain my positive evaluation of the work. Adding extending the discussion of of analyticity in $\beta^{-1}$ would be really helpful to make the paper more accessible to ML audience.

---

> > > ### Author Response · Authors · 2026-04-04
> > >
> > > We thank the reviewer again for the helpful suggestions and comments. We will certainly add a more rigorous derivation/discussion on the analyticity of free energy in $\beta^{-1}$.
> > >
> > > The reviewer chose (b) Partially resolved - I have follow-up questions for the authors, yet we did not see any follow-up questions in the above rebuttal acknowledgement. We are happy to discuss any further questions. Thanks again for the support and the positive rating.

---

### Decision · Program_Chairs · 2026-04-30

**Decision:**

Accept (regular)

**Comment:**

The paper demonstrates the scaling law $L\sim \tau^{-1/3}$ ($\tau$: optimization time) through a simple toy model based on one-layer softmax network with the cross-entropy loss. Under the aligned student ansatz, the 1/3-scaling law can be derived through the Taylor expansion of the free energy. Despite its simplicity, this 1/3 exponent can be empirically seen across a wide range of LLMs.

The three reviewers out of four regard the characterization of the scaling law through a simple toy model as a fresh insight and strong contribution. Nevertheless, one reviewer had a concern about this simplicity because the paper still primarily relies on a large amount of empirical measurements to 1/3 exponent.

During the author-reviewer discussions, there was a clarification on the meaning of the "universality" and why the law derived from a toy model holds widely, as follows:

> Because the argument of Taylor expansion of the CE loss and gradient descent dynamics will be the same for more complex situations. And if we keep the leading terms only, the scaling should not change.

This is a good enough clarification at this moment yet better appended in the subsequent version of the paper.

Also, the authors additionally demonstrated 1/3-scaling law for OLMo families. This is also a strong evidence to advocate for their contributions.

After addressing these points, the paper is ready for being accepted.